# Speg interactions that regulate the stability of excitation-contraction coupling protein complexes in triads and dyads

Chang Seok Lee[1,5], Sung Yun Jung [2,5], Rachel Sue Zhen Yee [1], Nadia H. Agha [1], Jin Hong[1], Ting Chang[1], Lyle W. Babcock[1], Jorie D. Fleischman[1], Benjamin Clayton[1], Amy D. Hanna[1], Christopher S. Ward [1], Denise Lanza [3], Ayrea E. Hurley[1], Pumin Zhang [4], Xander H. T. Wehrens[1], William R. Lagor [1], George G. Rodney [1] & Susan L. Hamilton [1✉]

Here we show that striated muscle preferentially expressed protein kinase α (Spegα) maintains cardiac function in hearts with Spegβ deficiency. Speg is required for stability of excitation-contraction coupling (ECC) complexes and interacts with esterase D (Esd), Cardiomyopathy-Associated Protein 5 (Cmya5), and Fibronectin Type III and SPRY Domain Containing 2 (Fsd2) in cardiac and skeletal muscle. Mice with a sequence encoding a V5/HA tag inserted into the first exon of the *Speg* gene (HA-Speg mice) display a >90% decrease in Spegβ but Spegα is expressed at ~50% of normal levels. Mice deficient in both Spegα and Speg β (Speg KO mice) develop a severe dilated cardiomyopathy and muscle weakness and atrophy, but HA-Speg mice display mild muscle weakness with no cardiac involvement. Spegα in HA-Speg mice suppresses $Ca^{2+}$ leak, proteolytic cleavage of Jph2, and disruption of transverse tubules. Despite it's low levels, HA-Spegβ immunoprecipitation identified Esd, Cmya5 and Fsd2 as Spegβ binding partners that localize to triads and dyads to stabilize ECC complexes. This study suggests that Spegα and Spegβ display functional redundancy, identifies Esd, Cmya5 and Fsd2 as components of both cardiac dyads and skeletal muscle triads and lays the groundwork for the identification of new therapeutic targets for centronuclear myopathy.

[1] Department of Integrative Physiology, Baylor College of Medicine, Houston, TX 77096, USA. [2] Department of Biochemistry, Baylor College of Medicine, Houston, TX 77096, USA. [3] Department of Molecular and Human Genetics, Baylor College of Medicine, Houston, TX 77096, USA. [4] The First Affiliated Hospital, Zhejiang University Medical School, Hangzhou, China. [5] These authors contributed equally: Chang Seok Lee, Sung Yun Jung. ✉email: susanh@bcm.edu

Centronuclear myopathies (CNM) are congenital myopathies characterized by centralized nuclei in myofibers and muscle weakness that ranges from mild to life-threatening[1]. CNM is associated with X-linked recessive mutations in *MTM1* (Myotubularin 1) and autosomal mutations in *DNM2* (Dynamin 2), *BIN1 (*Bridging Integrator 1), *RYR1* (Ryanodine Receptor 1), *CACNA1S (*Calcium Voltage-Gated Channel Subunit Alpha1 S), *TTN* (Titin), and *SPEG* (Striated Muscle Preferentially Expressed Protein Kinase)[1]. Bin1, Dnm2, and Mtm1 play critical roles in transverse tubule (t-tubule) formation and structure[1–9]. Mtm1, a lipid phosphatase, regulates the activities of PI3 kinase, Dnm2, and Bin1[10]. T-tubules in skeletal and cardiac muscle are invaginations of the sarcolemma that form dyads and triads with the sarcoplasmic reticulum in cardiac muscle and skeletal muscle, respectively. These structures are essential for excitation-contraction coupling (ECC) in striated muscle. RyR1, which is the skeletal muscle $Ca^{2+}$ release channel, and the dihydropyridine receptor (DHPR or $Ca_V1.1$), which is a voltage-dependent $Ca^{2+}$ channel (α1s subunit encoded by *CACNA1S*), are critical for ECC and localized to triads. Their counterparts in cardiac tissue, RyR2, and $Ca_V1.2$, are localized to dyads. Speg interacts with RyR2[11,12], Serca2a[13], and Jph2[12] in the heart and Mtm1[14], Dnm2[15], and desmin[10] in skeletal muscle, suggesting that Speg plays roles in both ECC and t-tubule organization. Collectively, these data suggest that CNM arises from the disruption of t-tubules and triadic/dyadic ECC complexes.

ECC is the process whereby an action potential activates $Ca_V1.2$ opening (cardiac) or causes a $Ca_V1.1$ conformational change (skeletal muscle) to trigger the opening of SR $Ca^{2+}$ release channels (RyR2 and RyR1, in heart and skeletal muscle, respectively), leading to muscle contraction[16]. The composition of the ECC complex has been studied for many years with proteins such as $Ca^{2+}$ release channels (RyR1 and RyR2)[17–19], junctophilins (Jph1 and 2)[20–23], triadin[24–26], junctin[27,28], FK506 binding proteins (Fkbp12 and Fkbp12.6)[29–31], Stac3[32–34], and calmodulin[35–38] identified as key components. More recently, Speg[6,10–12,14,15,39–42] and cardiomyopathy-associated 5 protein or myospryn (Cmya5)[43] have been identified as part of the ECC complex in cardiac and skeletal muscle, but their functional roles remain to be fully elucidated.

Speg is a member of the myosin light chain kinase (MLCK) protein family[1,6] with two kinase domains similar to those of the muscle giant protein obscurin. In striated muscle, Speg is expressed as two alternatively spliced isoforms: Spegα (250 kDa) and Spegβ (355 kDa), differing at the N-terminus due to different transcriptional initiation sites. Both Speg isoforms have multiple Ig and FnIII domains that allow them to interact with other proteins and possibly position their binding partners for phosphorylation. Agrawal et al.[14] found that Speg protein levels were severely depleted in patients with *Speg* mutations associated with CNM and suggested that Speg-deficient mice are good models for CNM. Huntoon et al.[39] created striated muscle-specific Speg-KO mice, which developed dilated cardiomyopathy and severe skeletal muscle myopathy. The skeletal muscle of these Speg-deficient mice displayed a reduced ability to generate force, abnormal triads, decreased $Ca_V1.1$-mediated $Ca^{2+}$ currents, decreased SR $Ca^{2+}$ release, altered ECC[39] and a deficiency in satellite cells[40]. Quick et al.[12] used tamoxifen-inducible cardiomyocyte-specific Speg-deficient mice to demonstrate that Speg deficiency in the heart causes heart failure (HF) associated with increased SR $Ca^{2+}$ spark frequency and disruption of the t-tubules. Agrawal and colleagues[39] demonstrated spatial variability in electrically evoked $Ca^{2+}$ transients in Speg-KO skeletal muscle fibers, reminiscent of Mtm1 deficiency[44]. Recently, Li et al.[15] found that decreasing the expression of Dnm2 in Speg-deficient mice improved skeletal but not cardiac muscle function in Speg-deficient mice.

Campbell et al.[11] identified S2367 as a Speg phosphorylation site on RyR2 and demonstrated that conversion of S2367 to an alanine to prevent phosphorylation promoted atrial fibrillation, while conversion to an aspartic acid to mimic phosphorylation prevented pacing-induced atrial fibrillation. Speg phosphorylation sites on RyR1 have not been identified, and the sequence, including S2367 in RyR2, is not conserved in RyR1.

Another likely target of Speg phosphorylation is Jph2, a protein that is essential for maintaining dyads[23] and t-tubules[45] in cardiac muscle. Abnormal t-tubules and disrupted dyads are associated with cardiac hypertrophy and heart failure[23]. Jph2 is the primary isoform expressed in cardiac muscle, and both Jph1 and Jph2 are expressed in skeletal muscle. The site of Speg phosphorylation of Jph2 has not yet been identified, but phosphorylation of this protein at structural motifs could have profoundly different functional outcomes. Junctophilins have eight N-terminal MORN (membrane occupation and recognition nexus) repeats, of which the first three interact with the voltage-dependent $Ca^{2+}$ channel ($Ca_V1.1$ or $Ca_V1.2$)[46]. The MORN repeats form a continuous β sheet and are followed by a long helix that serves as a backbone for the sheet. The C-terminal domain interacts with the SR membrane and, possibly, with RyRs[47]. Jph2 is cleaved by calpains in response to increased cytosolic $Ca^{2+}$ levels, and an N-terminal fragment of Jph2 in the heart translocates to the nucleus, where it serves as a stress-activated transcription factor[48–50]. A mutation in the joining domain between the 6th and 7th MORN domains causes disruption of the dyads[51]. This joining domain interacts with $Ca_V1.2$ to recruit it to the t-tubules[51]. Mutations and deficiencies in RyRs, Speg[11,12,39], Jph2[43,52], and Cmya5[43] (myospryn/TRIM76) cause abnormal t-tubules, aberrant SR $Ca^{2+}$ release, and cardiac and/or skeletal muscle dysfunction[43,52,53], but the details of how these proteins structurally and functionally interact remain to be fully elucidated.

In this manuscript, we describe a mouse model that expresses Spegβ with an HA-Tag (HA-Speg mice). These mice display a significant decrease in Spegβ and a lesser decrease in Spegα (not HA-tagged) protein levels. We compared the effects of Speg deficiencies in Speg-KO (MCK-Cre$^+$$Speg^{fl/fl}$) and HA-Speg mice on $Ca^{2+}$ spark frequency and t-tubule disruption to show that $Ca^{2+}$ leak occurs in areas of t-tubule disruption. We also evaluate the effects of Speg deficiency on the levels of ECC proteins, Jph2 fragmentation, and interactions among triadic and dyadic proteins and relate these changes to functional differences between HA-Speg and Speg-KO mice. We use HA-Speg mice to confirm RyRs and Jph2 as Speg-binding proteins and to identify new interacting proteins, including Esd, Cmya5 and Fsd2, in both heart and skeletal muscle. We use BioID with Fkbp12-BirA to show that these currently identified Speg-binding proteins localize to the triads in skeletal muscle and, using RyR and Jph2 immunoprecipitations, we confirm that Esd, Cmya5, and Fsd2 are Speg-dependent components of RyR/Jph2 triadic and dyadic complexes and that the absence of Speg leads to destabilization of triadic and dyadic ECC protein complexes. Our study shows that the severity of the disease associated with Speg deficiency correlates with t-tubule disruption, increased $Ca^{2+}$ spark frequency in regions of t-tubule disruption, decreased ECC proteins, Jph2 fragmentation, and disruption of ECC protein complexes, and that a deficiency in Spegβ is partially rescued by Spegα.

## Results

**Creation of HA-Speg mice and comparison with Speg-KO mice.** Because of difficulties encountered with Speg immunoprecipitation using several currently available Speg antibodies, we created a mouse model to identify Speg-binding partners. Using CRISPR/Cas9-mediated genome editing to insert a sequence

encoding a V5 (a 14 amino acid sequence from simian virus[54]) combined with a 3XHA tag (9 amino acid sequence from influenza virus hemagglutinin) (Supplementary Fig. 1) into the *Speg* gene, we created mice with the V5/HA-tagged Spegβ (HA-Speg mice) to use for identification of Speg-binding proteins. Since the sequence encoding the tag is inserted in the first exon of *Spegβ*, only Spegβ has the V5/HA-tag. The heterozygous mice were backcrossed with wild-type mice at least twice to reduce off-target effects of the editing, and then the heterozygous mice were crossed to produce homozygous HA-Spegβ mice. The insertion of the V5/HA-tag decreased Spegβ and Spegα protein levels in the *tibialis anterior* (TA) muscle of homozygous HA-Speg mice by $95.1 \pm 0.8\%$ (mean ± SD) and $50.9 \pm 8.7\%$, respectively (Fig. 1a), and, in the heart, Spegβ and Spegα levels were reduced by $82.1 \pm 4.2\%$ and $38.3 \pm 6.7\%$, respectively (Fig. 1c). Hence, we have created a mouse model of Spegβ deficiency with some sparing of Spegα. In contrast, MCK-Cre+*Speg^{fl/fl}* mice[11] (driven by the muscle creatine kinase promoter, which we designate as Speg-KO mice) display a >95% decrease in both Spegβ and Spegα in both skeletal muscle (Fig. 1b) and heart (Fig. 1d).

To elucidate the cause(s) of the reduction in Speg isoforms in HA-Speg mice, we measured relative mRNA levels for Spegβ, Spegα, and two splice variants of the *Speg* gene, Apeg, and Bpeg in skeletal and cardiac muscle. As can be seen in Fig. 1e, only the message for Spegβ was decreased in the muscle of HA-Speg mice and showed a tendency toward a decrease in the heart (Fig. 1f). mRNA levels for Spegα, Apeg, and Bpeg were not significantly altered. It seems unlikely that a 50% decrease in mRNA would lead to a 95% decrease in Spegβ levels. However, the 3xHA tag has been shown to cause increased proteolytic degradation of a variety of different tagged proteins[55], suggesting that some of the decrease in HA-Spegβ are likely due to protein turnover. Since Spegβ and Spegα have identical C-terminal sequences, we cannot use mass spectrometry to distinguish between the two isoforms using immunoprecipitation of Speg. However, in HA-Speg mice, only Spegβ has the HA-tag, but the HA antibody coimmunoprecipitates a second band with the MW of Spegα that is recognized by the Speg antibody but not HA antibody (Fig. 1g), suggesting that some Spegα is bound to Spegβ. The HA-tag on Spegβ may render Spegα complexed to HA-Spegβ vulnerable to turnover.

ECC protein levels have been reported to be reduced in the skeletal muscle of Speg-deficient mice[10], and, hence, quantitation of changes in ECC protein levels is needed to interpret functional and immunoprecipitation data. We compared the effects of Speg deficiency on RyR1 and Cacna1s (Ca$_V$1.1α1s) and RyR2 and Cacna1c (Ca$_V$1.2α1c) protein levels in the TA muscle and heart, respectively, of both HA-Speg and Speg-KO mice relative to controls (wild type, WT, for HA-Speg and *Speg^{fl/fl}* for the Speg-KO). RyR1 and Ca$_V$1.1α1 were reduced by $34 \pm 7\%$ and $50 \pm 7\%$, respectively, in the skeletal muscle of HA-Speg mice compared to control mice (Fig. 1h) and by $83 \pm 4\%$ and $65 \pm 9\%$, respectively, in the TA muscle of Speg-KO mice versus control mice (Fig. 1i). Neither RyR2 nor Ca$_V$1.2α1c were reduced in the hearts of either HA-Speg (Fig. 1j) or Speg-KO mice (Fig. 1k). Hence, while decreased ECC proteins may contribute to the altered skeletal muscle function, a different mechanism appears to underlie the cardiac phenotype.

## Functional consequences of Speg deficiency in Speg-KO and HA-Speg mice.
To evaluate the functional consequences of the reductions in Speg in HA-Speg compared to Speg-KO mice, we performed several functional tests on HA-Speg mice compared to control (WT) mice. Similar analyses were performed with the Speg-KO and control (*Speg^{fl/fl}*) mice. Neither male nor female HA-Speg mice developed dilated cardiomyopathy

(Supplementary Table 1), nor did they not die prematurely. Consistent with published results[14,39], both male and female Speg-KO mice developed a dilated cardiomyopathy (DCM) (Supplementary Table 1). These data suggest that: (1) smaller amounts of Spegβ in HA-Speg mice are adequate to support cardiac function, (2) Spegα can compensate for the loss of Spegβ and/or, (3) Spegα rather than Spegβ prevents the development of DCM. The HA-Speg male mice were slightly smaller than the age-matched WT (control) mice and had less lean mass (Supplementary Table 2). Both male and female Speg-KO mice displayed significant decreases in body weight, length, bone area, fat weight, lean weight, bone mineral concentration (BMC), and bone mineral density (BMD) (Supplementary Table 2).

Ex vivo force generation was reduced in the soleus (Fig. 2a) and the diaphragm (Supplementary Fig. 2a), but not in the *extensor digitorum longus* (EDL) (Fig. 2b) of HA-Speg mice. The force-frequency curves were significantly right-shifted by 4.1, 0.7, and 6.0 Hz for the soleus, EDL, and diaphragm muscles, respectively, of male HA-Speg mice compared to WT controls (Fig. 2a, b and Supplementary Fig. 2a). For comparison, muscles from male Speg-KO mice were more severely affected than those from HA-Speg mice, especially the EDL. Maximal force generation was decreased in the soleus (Fig. 2c), EDL (Fig. 2d), and diaphragm (Supplementary Fig. 2b) of the Speg-KO male mice. The force-frequency curves were also shifted to the right by an average of 17, 59, and 24 Hz in the soleus, EDL, and diaphragm muscles, respectively, of the Speg-KO male mice compared to controls. This rightward shift in the half-maximal stimulation frequency detected in all muscles in the Speg-KO mice is likely to be the result of the disruption or inhibition of ECC and/or t-tubules and triads.

Similar experiments were performed with female mice. Maximal force was not significantly decreased in the soleus (Supplementary Fig. 2c) or EDL (Supplementary Fig. 2d) of the female HA-Speg mice, but maximal force in the diaphragm of the female HA-Speg mice was significantly decreased compared to WT controls (Supplementary Fig. 2e). Maximal force was reduced in the soleus (Supplementary Fig. 2f), EDL (Supplementary Fig. 2g) and diaphragm (Supplementary Fig. 2h) of the female Speg-KO mice compared to controls. The half-maximal stimulation frequency was shifted by ~6 Hz in the diaphragm of the female HA-Speg mice and by an average of 26, 45, and 22 Hz for the soleus, EDL, and diaphragm muscles, respectively, of Speg-KO female mice compared to controls. Hence, Speg deficiency has similar effects in male and female mice.

We evaluated the effects of Speg deficiency in male HA-Speg and Speg-KO mice on the cross-sectional area (CSA) and fiber-type distribution of muscle fibers in the soleus and EDL using MyoSight[56]. We found no significant differences in CSA in Type I, IIa, or IIx in the soleus (Fig. 2e, i) or EDL (Fig. 2f, j) of the HA-Speg compared to controls. However, the CSA of IIx fibers was reduced in the soleus of Speg-KO mice compared to controls (Fig. 2g, k), and the CSA of all fiber types were dramatically reduced in the EDL (Fig. 2h, l). While there were no significant changes in fiber-type distribution in soleus (Fig. 2m) or EDL (Fig. 2n) of HA-Speg mice, there was a significant decrease in the % of IIx fibers in soleus (Fig. 2o) of Speg-KO mice. The EDL of Speg-KO mice displayed a decrease in the percentage of IIa fibers and an increase in the % of IIb fibers (Fig. 2p). CSA distributions for each type of fiber are shown in Supplementary Fig. 3. Based on these analyses, decreased muscle fiber size and fiber-type switching are major features of the Speg-KO but not HA-Speg mice. Also, our ex vivo force measurements and fiber typing studies show that fast twitch muscles are affected to a much greater extent by Speg deficiency than slow twitch muscles, and both male and female mice are affected in Speg-KO mice.

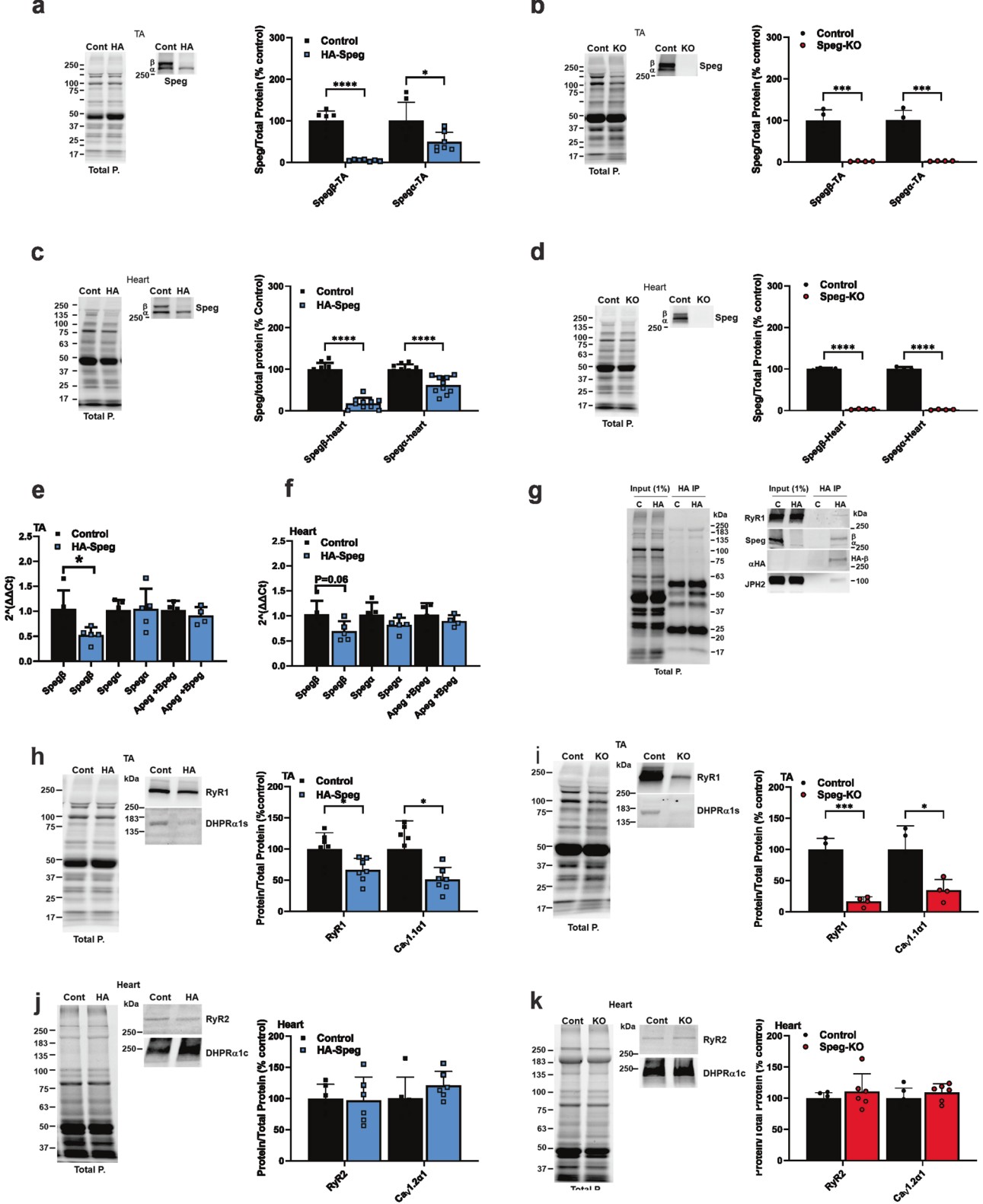

**Effects of Speg deficiency on SR Ca²⁺ leak and t-tubule structure**. Normally, RyR1 opening is tightly controlled by the dihydropyridine receptor (DHPR, Ca$_V$1.1) within the triads of fully differentiated skeletal muscle, and this control serves to suppress spontaneous Ca²⁺ release (Ca²⁺ sparks)[57]. Huntoon et al.[39] showed that Speg deficiency promoted spatial variability in macroscopic Ca²⁺ transients and disrupted triads in skeletal

muscle fibers. We examined the t-tubule structure in dissociated *flexor digitorum brevis* muscle (FDB) fibers from HA-Speg, Speg-KO, and control mice. There were areas of abnormal t-tubules in FDB fibers from HA-Speg mice (Fig. 3a, b), but the occurrence of these disruptions was rare. This is in sharp contrast to the FDB fibers from the Speg-KO mice (Fig. 3e, f), where most t-tubules were aberrant. Our findings in the Speg-KO are consistent with

**Fig. 1 Speg and ECC protein expression levels in the TA and cardiac muscle of male HA-Speg and Speg-KO mice. a** Spegβ and Spegα expression in the TA of HA-Speg mice (homozygous) compared to controls (WT) ($n = 7$). A representative western blot is shown on the left. Total protein (in all panels) was obtained using stain-free gels (Bio-Rad), and after transfer of the proteins to the PVDF membrane (Immobilon®-FL, Millipore), the PVDF membrane was imaged by using a ChemiDoc™ MP imaging system (Bio-Rad). **b** Spegβ and Spegα expression in the TA of Speg-KO mice (homozygous) compared to controls (*Speg*[fl/fl]) ($n = 4$ each). A representative western blot is shown on the left. **c** Spegβ and Spegα expression in hearts of HA-Speg mice compared to control mice ($n = 10$ each). A representative western blot is shown on the left. **d** Spegβ and Spegα expression in the hearts of Speg-KO mice (homozygous) compared to controls (*Speg*[fl/fl]) ($n = 4$ each). A representative western blot is shown on the left. **e** Expression of mRNA for Spegα, β, Apeg-1 + Bpeg in TA muscles of control ($n = 4$–5) and HA-tagged Speg mice ($n = 4$–5), respectively. *P* values are indicated as analyzed by Welch's *t* test. All statistical tests are two-sided. Data are represented as mean ± standard deviation. **f** Expression of mRNA for Spegβ, Spegα, Apeg-1, and Bpeg in hearts of control ($n = 4$–5) and HA-tagged Speg mice ($n = 4$–5), respectively. *P* values are indicated as analyzed by Welch's *t* test. All statistical tests are two-sided. **g** Immunoprecipitation with HA antibody of homogenate from HA-Speg and control mice. The figure shows western blot of total proteins and western bot with Speg antibody and HA antibody. **h** ECC protein levels in the TA of Speg-KO mice compared to controls with a representative western blot ($n = 7$ each). **i** ECC protein levels in TA of Speg-KO mice compared to controls with a representative western blot ($n = 4$ each). **j** ECC protein levels in the hearts of Speg-KO mice compared to controls with a representative western blot ($n = 6$ each). **k** ECC protein levels in hearts of Speg-KO mice compared to controls with a representative western blot ($n = 6$ each). All mice used for the data in this figure were male. Data are shown as the mean ± SD. ****$P < 0.0001$, ***$P < 0.001$, **$P < 0.01$, *$P < 0.05$.

the findings of Huntoon et al.[39], with most fibers from the Speg-KO having an abnormal t-tubule structure. T-tubule disorganization ranged from the absence of t-tubules within areas of the myofiber to t-tubule expansion/vacuolation. Using dual imaging of t-tubules (FM4-64) and $Ca^{2+}$ (Fluo-4), we found a significant increase in spontaneous $Ca^{2+}$ sparks in both HA-Speg (Fig. 3c) and Speg-KO fibers (Fig. 3g) and that the $Ca^{2+}$ sparks were located primarily in microdomains of t-tubule disruption in fibers from both HA-Speg (Fig. 3b) and Speg-KO (Fig. 3f). The appearance of $Ca^{2+}$ sparks in adult muscle is likely to be a sign of disruption of $Ca_V1.1$-RyR1 coupling. T-tubules in areas of spontaneous $Ca^{2+}$ sparks showed structural disorganization, with increased density of longitudinal tubules and decreased transverse tubules, regularity, and integrity (Fig. 3d, h). These findings raise the question of whether increased $Ca^{2+}$ sparks (due to altered phosphorylation of RyRs and/or altered interactions within the triadic ECC complex) activate calpain to cleave Jph2 and disrupt t-tubules or whether the disruption of the t-tubules by the loss of another Speg-dependent pathway (possibly the loss of Jph2 phosphorylation or its interactions with other ECC proteins) alters the structure of the triads/dyads, causing an increase in $Ca^{2+}$ sparks. The two events together are likely to drive a feed-forward, amplifying effect of Speg deficiency with neither change alone being adequate to drive the triadic disruption.

**Jph2 fragmentation in HA-Speg and Speg-KO mice.** Since HA-Speg mice have a much milder phenotype than Speg-KO mice, a major goal of this study was to compare HA-Speg and Speg-KO mice to find potential pathways where the level of Spegα+Spegβ is critical to the phenotype. Since Jph2 and its calpain-mediated cleavage play a critical role in t-tubule and triad structure[5,39], we evaluated the extent of Jph2 cleavage in the two mouse models to determine if Jph2 levels and cleavage were different in the two mouse models of Speg deficiency.

Jph2 levels in the heart are downregulated by cardiac stress[58–60], and the decrease is, at least partially, due to calpain cleavage of Jph2 into fragments[49,50,61,62]. Junctophilins are required for the maintenance of the structure of skeletal muscle triads and cardiac dyads[63]. Based on the Jph2 sequence, Jph2 should have a molecular weight (MW) of 75 kDa, but it migrates on sodium dodecyl sulfate (SDS) polyacrylamide gel electrophoresis (PAGE) with an apparent MW of ~101 kDa (Fig. 4a, green Ab J2-431-680). Jph1 (MW of 72 kDa) migrates in SDS-PAGE with an apparent MW of 87 kDa (Fig. 4a, red J1-Ab 559-572). Full-length Jph2 is labeled as band a in Fig. 4b, d, f, h. The anomalous migration of junctophilins on SDS gels could be due to posttranslational modifications such as palmitoylation[64],

interactions with other proteins that are not disrupted by SDS, or an unusual conformation of the partially unfolded Jph. The protein levels of full-length Jph2 were reduced by 28 ± 10% in the tibialis anterior (TA) muscle of HA-Speg mice (Fig. 4b, c) and by 45 ± 9% in the TA muscle of Speg-KO mice (Fig. 4d, e). Jph2 was reduced by 32 ± 20% in the hearts of HA-Speg mice (Fig. 4f, g) and by 55 ± 6% in the hearts of Speg-KO mice (Fig. 4h, i). Hence, both models of Speg deficiency display a decrease in skeletal muscle and heart levels of Jph2, with the decreases being greater in Speg-KO mice.

The anomalous migration of both Jph2 and Jph1 makes it extremely difficult to estimate the size of the Jph2 fragments and/or identify calpain cleavage sites. Weninger et al.[62] identified calpain cleavage sites on recombinant Jph2 (which migrated as a >100 kDa protein) and ordered the sites sequentially using increasing concentrations of calpain[62]. Sequentially, calpain was suggested to cleave Jph: (1) after arginine 565 (in agreement with Guo et al.[50]), (2) after serine 164, (3) between amino acids 235 and 297, and (4) between 590 and 612[62]. While this study identified sites that can be cleaved by calpain in the context of recombinant Jph2, calpain cleavage sites on Jph2 in cardiac or skeletal muscle could be masked by protein-protein interactions or Jph2 posttranslational modifications (e.g., phosphorylation, palmitoylation). Guo et al.[65] identified calpain cleavage sites at Val-155/Arg-156, Leu-201/Leu-202, and Arg-565/Thr-566.

Our goal was to determine if Speg deficiencies in our two mouse models had similar effects on Jph2 cleavage. To identify Jph2 fragments in TA and cardiac muscle homogenates HA-Speg and Speg-KO mice compared to their controls using western blotting, we employed three validated sequence-specific antibodies (Fig. 4j): (1) an antibody to Jph2 amino acids 1–50 (**J2-1-50**). (2) an antibody to Jph2 amino acids 431–680 (J2-431-680), and (3) an antibody to Jph2 amino acids 565–580 (J2-565-580), which is immediately after the first putative calpain cleavage site[62]. The fragmentation patterns in western blot panels using these antibodies for the TA of HA-Speg mice are shown in Fig. 4b, c. The Jph2 fragments in the Speg TA of Speg-KO mice are shown in Fig. 4d, e. The Jph2 fragments identified in Fig. 4d as bands a–d are explained in Fig. 4j and quantified in Fig. 4e. Full-length Jph2 is band a in Fig. 4 and is recognized by all three antibodies. While full-length Jph2 (band a) was decreased in the TA muscle of both the HA-Speg and Speg-KO mice, none of the proteolytic fragments in the TA of HA-Speg mice increased compared to controls. Instead, the 91 kDa (band b) and the 28 kDa (band d) fragments actually displayed small decreases (Fig. 4c, d). Three bands were increased in the TA muscle of Speg-KO mice including a 91 kDa (band b), a 70 kDa (band c),

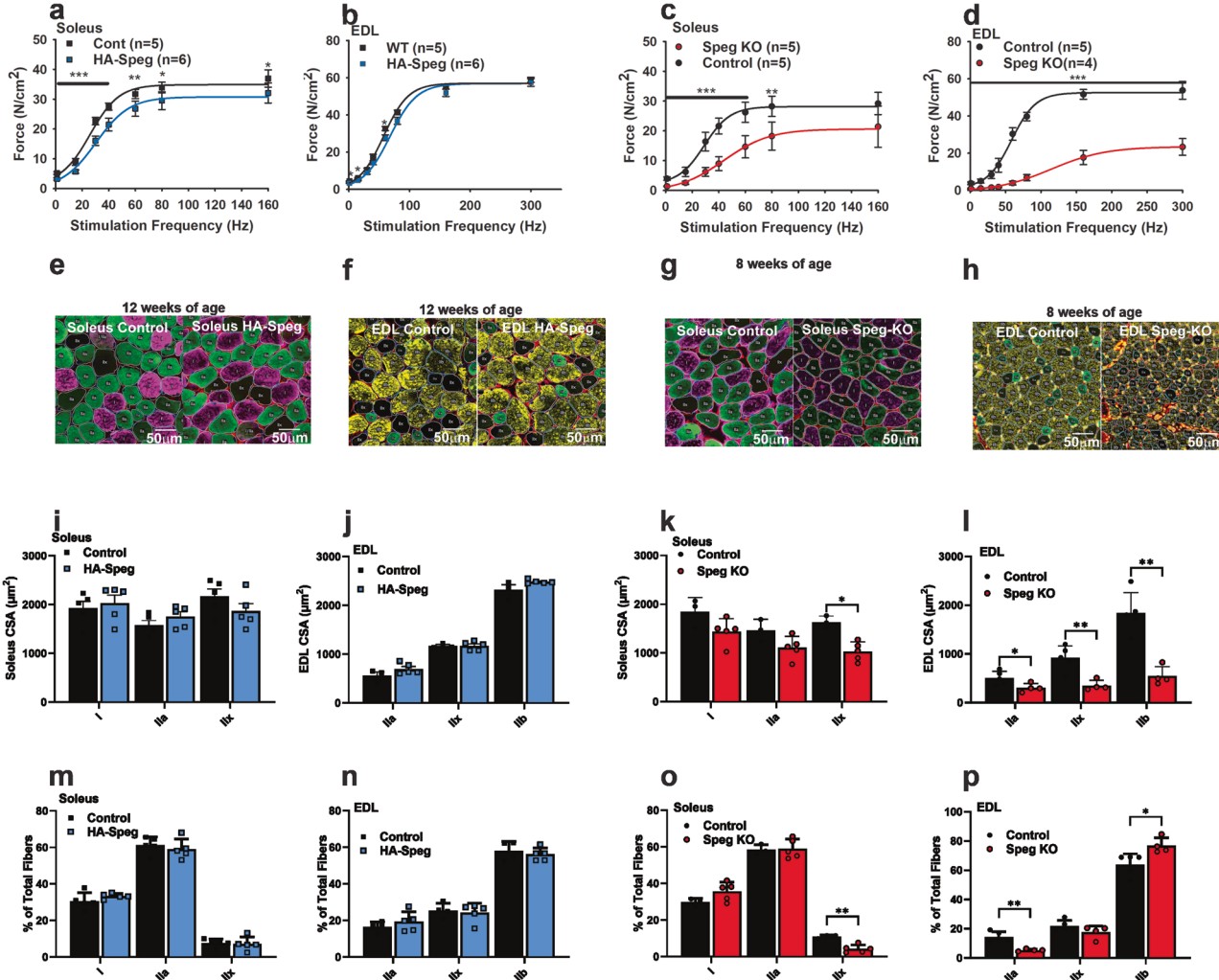

**Fig. 2 Functional consequences of Speg deficiency.** Skeletal muscle function in HA-Speg (14 weeks), WT (13 weeks), Speg-KO (8–9 weeks), and *Speg^fl/fl* (8–9 weeks) male mice were assessed using the indicated skeletal muscle functional or structural measurements. All data in this figure were obtained with male mice. Data on the diaphragm and separate data on female mice of each genotype are provided in the Supplementary Data Fig. 2. The Speg-KO mice used were younger than HA-Speg mice because they become extremely sick at ~10 weeks of age and often do not survive the echoes. **a** Force-frequency data for control (WT) and HA-Speg soleus. **b** Force-frequency data for the control (WT) and HA-Speg EDL. **c** Force-frequency data for control and Speg-KO soleus. **d** Force-frequency data for control and Speg-KO EDLs. **e** Fiber-type-specific staining of a representative cross-section from the soleus of control and HA-Speg mice. **f** Fiber-type-specific staining of a representative cross-section from the EDL of control and HA-Speg mice. **g** Fiber-type-specific staining of a representative cross-section from the soleus of control and Speg-KO mice. **h** Fiber-type-specific staining of a representative cross-section from the EDL of control and Speg-KO mice. **i** Average CSA of different fiber types in the soleus of control and HA-Speg mice. **j** Average CSA of different fiber types in the EDL of control and HA-Speg mice. **k** Average CSA of different fiber types in the soleus of control and Speg-KO mice. **l** Average CSA of different fiber types in the EDL of control and Speg-KO mice. **m** Fiber-type distribution in the soleus of control and HA-Speg mice. **n** Fiber-type distribution in the EDL of control and HA-Speg mice. **o** Fiber- type distribution in the soleus of control and Speg-KO mice. **p** Fiber-type distribution in the EDL of control and Speg-KO mice. Data are plotted as the mean ± SD. ****$P < 0.0001$, ***$P < 0.001$, **$P < 0.01$, and *$P < 0.05$.

and a 28 kDa (band d) (Fig. 4d, e). The 91 kDa fragment (band b) was recognized by J2-431-680 and J2-565-580, but not by the N-terminal (J2-1-50) antibody, suggesting a Speg-protected cleavage site close to the N-terminus. The N-terminal antibody also recognized an 85 kDa band, but this band was not detected by any other Jph2 antibody and was not different between controls and Speg-KO muscle and is, hence, likely to represent a nonspecific band for the N-terminal antibody. The 70 kDa fragment (band c) was recognized by the N-terminal antibody (J2-1-50) and the J2-431-680 antibody, but not by the J2-565-580 antibody suggesting the removal of a C-terminal fragment due to cleavage somewhere before amino acid 565. The 28 kDa fragment (band d) was recognized by J2-431-680 and J2-565-580, but not by the N-terminal antibody suggesting that it is a C-terminal

fragment with a cleavage somewhere before the S565 site. Band d may be generated by the same cleavage that generated the 70 kDa (band c) N-terminal fragment. We conclude that a large decrease in Speg allows Jph2 to be calpain cleaved in skeletal muscle at sites in both the N-and C-terminal regions of Jph2. Surprisingly, despite a decrease in full-length Jph2 in the skeletal muscle of HA-Speg mice, the fragments that increase in Speg-KO mice show small decreases in HA-Speg mice. This is likely to reflect different mechanisms of Jph2 degradation in the two mice but suggests that Jph2 fragmentation alone cannot explain the difference in skeletal muscle phenotypes in the two Speg-deficient mice. The N-terminal Jph2 fragment (band c) may function as transcriptional regulators[48–50], was only in the Speg-KO skeletal muscle. Hence, the accumulation of a Jph2

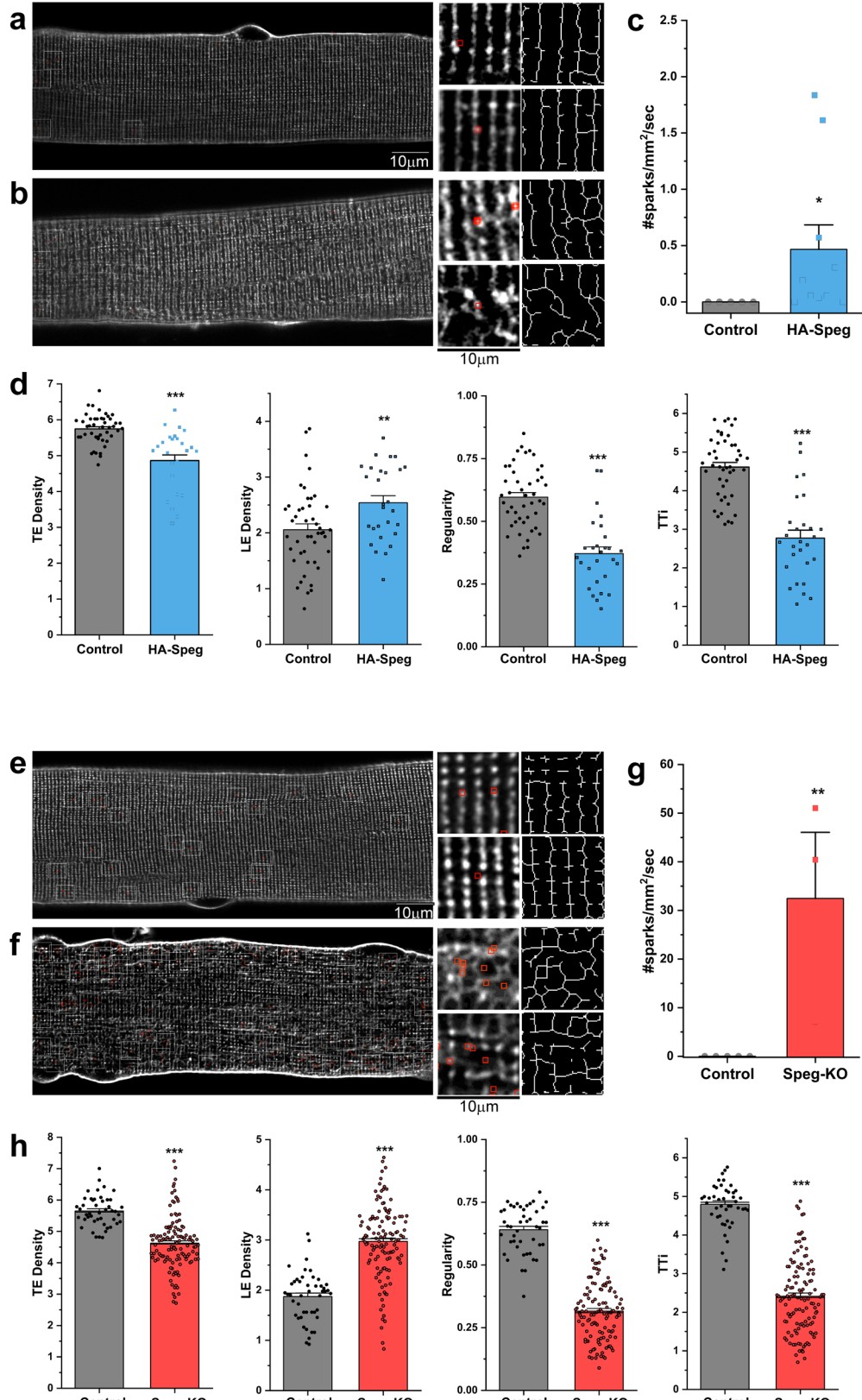

N-terminal fragment is another factor that could contribute to differences in the phenotype of the Speg-KO and HA-Speg mice.

We also examined the Jph2 fragmentation patterns in the hearts of HA-Speg (Fig. 4f, g) and Speg-KO (Fig. 4h, i) mice. We did not detect the C-terminal 28-kDa fragment in the heart, possibly indicating that is rapidly proteolyzed to smaller fragments in the heart. Similar to skeletal muscle, the 91-kDa and 70-kDa fragments increased in the hearts of Speg-KO mice compared to controls and showed small decreases in the hearts of HA-Speg mice. The 70-kDa fragment is likely to be the N-terminal fragment identified to function as a transcription factor[48–50].

**Fig. 3 Effects of Speg deficiencies on T-tubule structure and SR Ca$^{2+}$ leak (sparks).** FDB fibers were isolated from male mice as described in Methods. We used dual imaging of t-tubules (FM4-64) and Ca$^{2+}$ (Fluo-4) to find areas of T-tubule disruption and SR Ca$^{2+}$ leak. **a, b** T-tubule organization (raw and skeletonized) and spark localization (red squares) in FDB fibers from control (**a**) and HA-Speg (**b**) mice. **c** Spontaneous Ca$^{2+}$ spark frequency in control ($N_{fibers}=21$, $N_{animals}=5$) and HA-Speg fibers ($N_{fibers}=28$, $N_{animals}=6$). **d** Analysis of T-tubule organization in control and HA-Speg fibers. **e, f** T-tubule organization (raw and skeletonized) and spark localization (red squares) in FDB fibers from control (**e**) and Speg-KO (**f**) mice. **g** Spontaneous Ca$^{2+}$ spark frequency in control ($N_{fibers}=20$, $N_{animals}=5$) and Speg-KO fibers ($N_{fibers}=15$, $N_{animals}=3$). **h** Analysis of T-tubule organization in control and Speg-KO fibers. TE is the density of transverse elements; LE is the density of longitudinal elements. Regularity is the organization or spacing and is the magnitude of the major frequency derived from the FFT of the image. TTi is the t-tubule integrity, which considers both the regularity and density of the t-tubules. Data are plotted as the mean ± SD, ***$P < 0.001$, **$P < 0.01$, and *$P < 0.05$.

Using a Jph1 antibody to a C-terminal region (J1-554-572), we detected 57 kDa, 19 kDa, and 18 kDa fragments in both the TAs of WT and Speg-KOs, but none of the fragments increased in the Speg-KOs (Supplementary Fig. 4a, b), and this was not further investigated. For comparison, the Jph1 blots in the TA of HA-Speg mice are shown in Supplementary Fig. 4c, d.

**HA-Speg-binding proteins in triads.** Despite the decrease in HA-Spegβ levels in the skeletal muscle of the homozygous HA-Speg mice, the amount of remaining HA-Spegβ was adequate to immunoprecipitate HA-Spegβ binding proteins in both tissues. Nonspecific interactions were defined with HA antibody pull-downs from the muscle of WT mice. The data for this IP and all of the other IPs described below are presented in Supplementary Data 1, with different tabs for each IP (SD1-12). Proteomic data obtained from immunoprecipitations (IPs) with HA antibody from gastrocnemius muscles of HA-Speg and WT (nonspecific, ns) mice were analyzed by first removing all proteins that did not show >10×-fold purification (a minimal requirement for immune precipitation), and then we used a Holm–Šídáks multiple comparison test and/or false discovery rate (FDR)[66]. A plot of false discovery rate is shown in Fig. 5a. The major cytosolic and ER/SR proteins in the HA-Speg pulldown were Speg, Esd (Esterase D, a serine hydrolase), Fsd2 (FN3/SPRY domain protein or minispryn), Cmya5 (Cardiomyopathy-Associated 5 protein or myospryn), Jph2 and Jph1. We do not consistently detect Serca, Dnm2, Mtm1, or desmin as specific in the HA-Speg IPs. We hypothesize that this could be due to: (1) HA-tag interference with these interactions, (2) low-affinity interactions and these proteins are lost during the washes, (3) differences in HA-Speg turnover in different cellular compartments compared to triads, leading to an enrichment of triadic proteins in the HA-Speg IPs and/or, (4) these previously identified Speg-binding proteins are binding exclusively to Spegα.

Cmya5 is critical for the formation of cardiac dyads[15] and interacts with Speg and RyR2[15]. Since Fsd2 and Esd have not been previously shown to be Speg-binding proteins, we used antibodies to these proteins and western blotting of the immunoprecipitates to confirm the presence of Speg in the IPs (Supplementary Fig. 5a, b). We found Esd and Speg specifically in the Fsd2 IPs and Fsd2 and Speg specifically in the Esd IPs, suggesting that these two proteins form a complex with Speg. To confirm these interactions, we immunoprecipitated Fsd2 and evaluated the immunoprecipitated proteins by mass spectrometry. A heatmap of the mass spec values (iBaq) for specific proteins (20x over IgG and $P < 0.01$) is shown in Fig. 5b. Also shown in this panel are the effects of Speg deficiency (Speg-KO). Note that Speg deficiency decreases the recovery of Fsd2 in the IP suggesting that, in this analysis, the effects of Speg could, at least partially, reflect the decrease in Fsd2. However, this is also a second measure of specificity. These data (ECC proteins indicated by red arrows) demonstrate that: (a) the major Fsd2 binding proteins are Tfg (Trafficking From ER To Golgi Regulator), Esd, Speg and Cmya5, (b) RyR1, Jph2, and Jph1 are also found in the

Fsd2 IP, and (c) Esd, Fsd2, Speg, Cmya5, RyR1, Jph2, and Jph1 are all significantly decreased in the Fsd2 IP from Speg-KO mice. To account for the loss in Fsd2 and the parallel loss in binding proteins, we normalized each proteins to the amount of Fsd2 in the IP and compared the Speg-KO samples to their controls (Fig. 5c). While Esd did not significantly decrease (suggesting it directly binds to Fsd2), Speg, Cmya5, RyR1, Jph1 and jph2 were effectively removed from the IP by the Speg deficiency. To determine if the decreased Fsd2 in the Fsd2 IP from the muscle of Speg-KO is due to decrease Fsd2 protein levels, we performed western blots for Fsd2 and Esd in TA homogenates from the controls and Speg-KO mice (Fig. 5d, e). Both Fsd2 and Esd protein levels were decreased in muscle homogenates from Speg-KO (Fig. 5d) and HA-Speg mice (Fig. 5e). Tfg was not altered by Speg deficiency but its presence in the Fsd2 IP may reflect a role of Fsd2 in vesicular trafficking, a possibility that requires further study.

The presence of Esd, Fsd2, and Cmya5 in the HA-Speg IP could reflect Speg interactions outside of triads/dyads. To address this, we used a BioID approach[67] to identify triadic proteins close to the immunophilin Fkbp12 binding site on RyR1. Fkbp12 binds with high affinity and specificity to RyR1 to stabilize the closed state of channel[68]. Immunophilins, such as Fkbp12 and Fkbp12.6, are often used to purify RyRs for high-resolution cryoelectron microscopy (cryoEM) because of their high affinity and specificity for these channels[69–71]. A construct to express Fkbp12-BirA (a biotinylating enzyme) was packaged into an AAV virus and injected into the gastrocnemius muscle of Fkbp12-deficient mice[31] (MCK-Cre$^+$Fkbp12$^{fl/fl}$). A construct to express GFP-BirA in AAV was used as a control and injected into the contralateral leg. Mice ($n = 5$) were injected with biotin for 2 consecutive days prior to euthanasia at 3 weeks after the initial BirA virus injection, and the gastrocnemius muscle was isolated and homogenized. Biotinylated proteins were purified by streptavidin beads and identified by mass spectrometry. The specific biotinylated proteins were identified either as having a 10× fold purification over GFP-BirA and $P < 0.01$ or using false discovery rate, as shown in Fig. 5f. Near neighbors of FKBP12 in the triad included RyR1, Speg, Jph1, Jph2, Esd, Fsd2 and Cmya5. We also detected Ppp1r9b (Protein Phosphatase 1 Regulatory Subunit 9B) and Prpf4b (Pre-MRNA Processing Factor 4B, which is also a serine/threonine kinase). Ppp1r9b has been previously shown to bind to RyR2[72]. The significance, if any, of the interaction of Fkbp12 with Prpf4b remains to be determined. The biotinylation of Esd, Fsd2, and Cmya5 again suggests that these proteins are near the ECC apparatus in the triads. We did not detect specific labeling of triadin, calmodulin, PKA, or any of the subunits of Ca$_V$1.1, suggesting that these triadic proteins are not as close to the Fkbp12 binding site on RyR1 as the proteins in Fig. 5f.

**RyR1- and Jph2-binding proteins from skeletal muscle from control, Speg KO, and HA-Speg mice.** Since RyRs and Jph2 are likely interacting partners of Speg[11,12,39], we immunoprecipitated

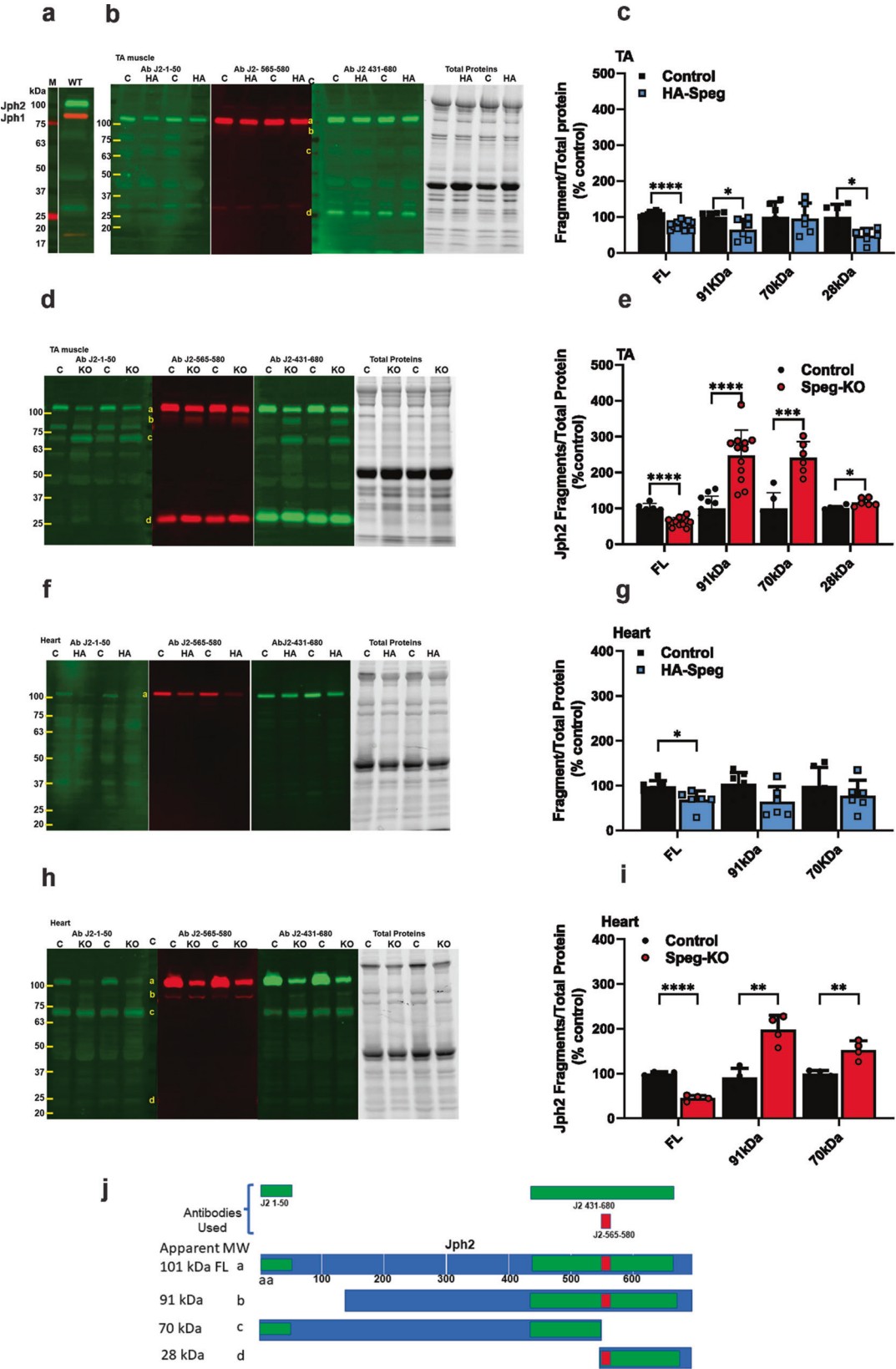

RyR1 and Jph2 from skeletal muscle (gastrocnemius) of *Speg*$^{fl/fl}$ (control) and Speg-KO mice to identify their binding proteins. In the proteomic experiments in Figs. 5 and 6, male mice were used to limit the number of mice since Speg deficiency had similar effects in male and female mice in skeletal muscle and heart. Both RyR1 and Jph2 IPs were performed in 2 or more independent

experiments, and each experiment used 1–3 control and 1–3 Speg-KO mice (*n* numbers for each experiment are provided in figure legends). Proteins in the IPs were identified by mass spectrometry. Datasets from each type of IP were pooled, and specific interacting proteins were identified by a 20X (RyR1) or 100x (Jph2) purification over IgG controls and a $P < 0.01$.

**Fig. 4 Effect of Speg deficiency on Jph fragmentation. a** Western blot for Jph2 and Jph1 in the TA of WT mice. Red Jph1. Green Jph2. **b** Western blot for Jph2 fragments in the TA of control and HA-Speg mice stained with three different antibodies. The antibodies used in the blots are indicated in panel j (see Supplementary Table 3). The far-right panel in (**b, d, f, h**) is the total protein. **c** Analyses of fragments in TA of control and HA-Speg mice ($n = 6$–13). **d** Western blot for Jph2 fragments in the TA of Speg-KO and control mice. **e** Analysis of Jph2 fragments in TA control and Speg-KO mice ($n = 6$–13). **f** Western blot for Jph2 fragments in hearts of control and HA-Speg mice. **g** Analyses of fragments in control and HA-Speg hearts ($n = 6$). **h** Western blot for Jph2 fragments in control and Speg-KO hearts. **i** Analysis of Jph2 fragments in hearts of control and Speg-KO mice ($n = 4$). **j** Diagram of antibody binding sites and possible calpain-mediated cleavage sites. The tissues used in this experiment were all from male mice are FL- full-length Jph2, **b**—91 kDa, **c**—70 kDa, **d**—28 kDa. Data are plotted as the mean ± SD. ****$P < 0.0001$, ***$P < 0.001$, **$P < 0.01$, and *$P < 0.05$.

To compare proteins in the IPs of controls to those in the Speg-KO, all samples were normalized to the amount of Jph2 or RyR1 in the IP and batch-corrected (compared to control IPs for that specific experiment) to assess the effects of the Speg deficiency.

We found 21 proteins in the RyR1 IP that met the criteria of specific ($P < 0.01$, 20× enrichment over control IgG) in the immunoprecipitate. A heatmap of the "specific" proteins is shown in Fig. 5g. The red arrows indicate the proteins that are different in the IPs from Speg-KO mice. Many of the specific proteins are known RyR1 binding proteins, including Fkbp12, Casq1, Speg, Jph1, Jph2, Triadin, and CaMKIIα and γ. Among the other proteins in the RyR1 IP, we find both Esd and Cmya5. Fsd2 is also present but is more variable than Esd and Cmya5 and does not reach a $P < 0.01$. Hence, at least two of the proteins identified in Fig. 5a, f are also in RyR1 IPs, again supporting the hypothesis that these proteins are triadic proteins. The significance (if any) of the presence of other proteins shown in the heatmap in Fig. 5g in the RyR1 IP is not yet known. RyR3 was also found in the RyR1 IP and is recognized by the same antibody as RyR1. We normalized each sample to the amount of RyR2 in that sample and then normalized to the mean of the control for that specific experiment. Specific proteins and proteins that are significantly different in the RyR1 IPs from Speg-KO mice are plotted in Fig. 5h. We found reductions ($P < 0.01$) in Casq1, Speg, obscurin (Obscn), Jph2, Jph1 and Cmya5 in the IPs from Speg-KO mice. For comparison, the ECC proteins reduced in the RyR1 IPs in the muscle of HA-Speg mice are shown in Fig. 5i. These data suggest that Speg interacts with RyR1 and/or an RyR1 binding protein and a major decrease in Speg causes dissociation of Casq1, Jph1 and 2, obscurin, Esd, and Cmya5 from RyR1. A curious finding in the RyR1 IPs was the presence of obscurin, a muscle giant protein that, like Speg, is a dual kinase of the MLCK family. We did not detect obscurin in either the HA-Speg IP (Fig. 5a) or the Fkbp12 BioID (Fig. 5f) experiment. While it is possible that the Speg antibody has affinity for obscurin, this would not explain the decrease in obscurin in the Speg-deficient mice. Obscurin links the sarcomere with the SR and interacts with titin and myomesin[73]. The deficiency in obscurin could contribute to the phenotype of Speg-KO mice, but mice deficient in obscurin display only a mild myopathy with centralized nuclei[74].

We immunoprecipitated Jph2 from the gastrocnemius muscles of control and Speg-KO mice. A heatmap showing "specific proteins" and proteins reduced in amount in the IPs by Speg deficiency is shown in Fig. 5j. The red arrows show proteins that are significantly different in the JPH2 IPs from Speg-KO mice and include Speg, Esd, Obscn. Specific proteins ($P < 0.01$, >100 purification over ns) included Jph2, Speg, RyR1, Asph (junctin), CaMKII (α, γ and δ), Esd, Cmya5, and Obscn. The immunoprecipitates of Jph2 from skeletal muscle had multiple ribosomal proteins that were removed for presentation purposes. We also performed a targeted comparison of ECC proteins in IPs from both Speg-KO (Fig. 5k) and HA-Speg (Fig. 5l) mice using the normalized and batch-corrected values (calculated as % control for each independent IP). The proteins in the control Jph2 IP were compared to those in Speg-KO IPs. As can be seen in

Fig. 5k, l, Speg deficiency in both mouse models decreased Speg, Fkbp12, RyR1, and Obscn in the Jph2 IPs. Cacna1s was significantly decreased in the Speg-KO but did not reach significance in HA-Speg IPs. The extent of reductions in Casq1, Speg and Obscn were greater in Speg-KO mice than in HA-Speg mice. We also compared the levels of Esd, Cmya5 and Fsd2 in the Jph2 IPs from Speg-KO (Fig. 5m) and HA-Speg (Fig. 5n) mice and found that the levels of these proteins were dramatically reduced in both models of Speg deficiency. A summary of the interactions of Speg and the identified Speg-binding proteins is shown in the model in Fig. 5o.

**Speg, RyR2, and Jph2-binding proteins in cardiac muscle.** The major Speg-binding proteins from the HA-Speg IPs from the heart were Speg, RyR2, Esd, Jph2, Cmya5 and Fsd2 (Fig. 6a). Cmya5 was previously identified as an RyR2 binding protein[75]. Hence, in addition to confirming Jph2 and RyRs as Speg-binding proteins, we identified 3 Speg-interacting proteins, Esd (esterase D), Fsd2, and Cmya5, in both skeletal and cardiac muscle. Similar to the situation in skeletal muscle, Speg deficiency in hearts of both Speg-KO (Fig. 6b) and HA-Speg (Fig. 6c) mice led to decreases in Esd and Fsd2 in cardiac muscle homogenates. The extent of these decreases was similar in the two mouse models.

We performed RyR2 IPs from hearts of control ($Speg^{fl/fl}$) and Speg-KO mice. A heatmap for the specific proteins in the RyR2 IP from control and Speg-KO hearts is shown in Fig. 6d. The red arrow represents the proteins that are significantly different in the RyR2 IP from hearts of Speg-KO mice and include RyR2, Speg, Jph2 and Fsd2. We found 11 "specific" proteins in the RyR2 IP, including RyR2, phospholamban, BC048546 (Alpha-2-Macroglobulin-like protein, a protease inhibitor), Speg, Jph2, Esd, CaMKIIδ, Ppapdc3 (an inactive sphingosine-1-phosphate phosphatase), Cmya5, Nfs1 (cysteine desulfurase) and Fsd2. We normalized each protein in the IP to the amount of RyR2 in the IP and then batch-corrected for each of the three independent experiments. Five proteins in the RyR2 IP from controls were significantly different in the IPs from hearts of Speg-KO (Fig. 6e), including Speg, Jph2, Esd, Cmya5, and Fsd2. Immunoprecipitations of RyR2 from the heart of HA-Speg mice were similar to those in the Speg-KO (Fig. 6f). Speg, Jph2, Esd, Cmya5 and Fsd2 were all reduced by Speg deficiency. BC048536 was reduced in the Speg-KO but was not reproducible in the HA-Speg.

We immunoprecipitated Jph2 from the hearts of controls, Speg-KO mice and HA-Speg mice. A heatmap of the values of a subset of proteins designated as specific is shown in Fig. 6g. The red arrows show values that are significantly different in the Speg-KO heart. Twenty-one proteins (ribosomal proteins removed) were identified as "specific" ($P < 0.01$ and 100× purification over IgG controls). We compared known ECC proteins in the IPs of the Speg-KO (Fig. 6h) and HA-Speg (Fig. 6i). Speg, RyR2, Casq2, Asph, and Trdn were all decreased in the Jph2 IPs from Speg-KO mice and Cacna1c was increased. Similar trends were seen in HA-Speg mice, but the extent of the decrease was less (Fig. 6i). Cacna1c was increased in the Jph2 IPs from HA-Speg mice. This is a major

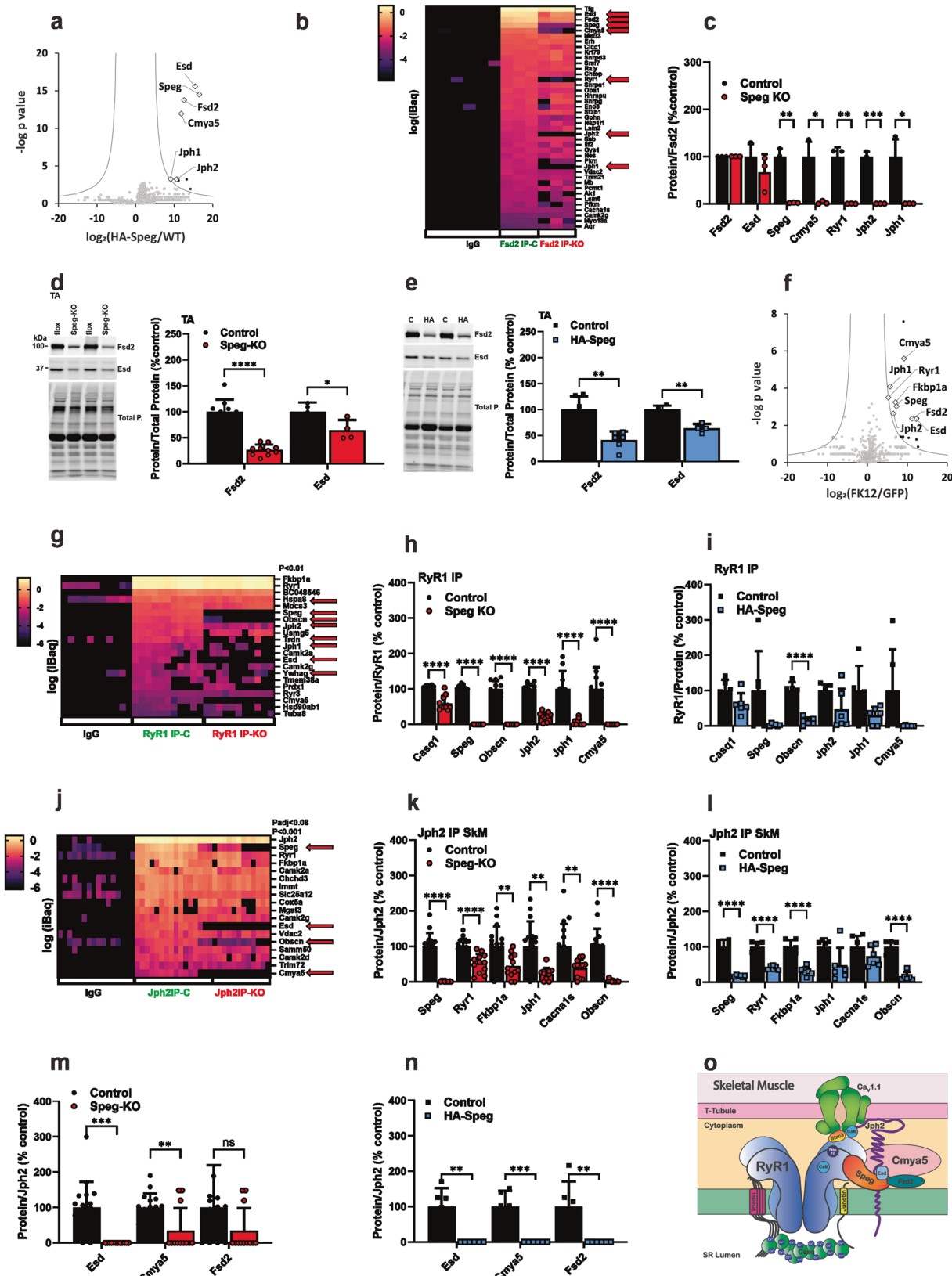

difference from skeletal muscle, where the interaction of Jph2 with the voltage-dependent $Ca^{2+}$ channel is decreased rather than increased by Speg deficiency. We also evaluated the levels of the three identified proteins and demonstrated that Esd, Cmya5 and Fsd2 were essentially absent in the Jph2 IPs from the hearts of both Speg-KO (Fig. 6j) and HA-Speg (Fig. 6k) mice. Since junctin binds

Casq, loss of these two proteins from the ECC complex in dyads may contribute to the more severe phenotype in Speg-KO mice. Immunoprecipitations of RyR2 and Jph2 from hearts of control and Speg-deficient mice suggest that the absence of Speg decreases the interactions among dyadic proteins and, similar to the situation in skeletal muscle, totally removes Esd, Cmya5, and Fsd2 from the

**Fig. 5 Speg-binding proteins and effects of Speg deficiency in skeletal muscle. a** Proteins immunoprecipitated from the gastrocnemius muscle of HA-Speg mice using antibodies against the HA-tag and identified by mass spectrometry ($n = 6$ for both the HA-IP from WT mice (nonspecific, ns) and the IP from HA-Speg mice). **b** Heatmap of immunoprecipitation of Fsd2 from Control and Speg-KO mice. ($n = 3$ for each). Also shown are IgG controls for nonspecific binding. **c** Analysis of the effects of Speg deficiency on Fsd2 binding proteins. **d** Effects of Speg deficiency on the levels of Fsd ($n = 10$) and Esd ($n = 4$) in homogenates from TA muscle of Speg-KO mice. A representative western blot is shown on the left. **e** Effects of Speg deficiency on the levels of Fsd2 ($n = 6$) and Esd ($n = 4$) in homogenates from TA muscle of HA-Speg mice. A representative western blot is shown on the left. **f** Proteins biotinylated by FKBP12-BirA compared to GFP-BirA. Proteins were purified with streptavidin beads and identified by mass spectrometry ($n = 5$ for each). **g** Heatmap of specific proteins in the RyR1 IP from the gastrocnemius muscle of control and Speg-KO mice ($n = 11$ for each). Nonspecific is shown as IgG. **h** RyR1 binding proteins in gastrocnemius muscle that are reduced in the RyR1 IP from Speg-KO mice ($n = 11$ for each). **i** RyR1 binding proteins in gastrocnemius muscle that are reduced in the RyR1 IP from HA-Speg mice ($n = 6$ for each). **j** Heatmap of specific proteins in the Jph2 IP from the gastrocnemius muscle of control and Speg-KO mice ($n = 14$ for control, $n = 12$ for Speg-KO). Also shown are IgG controls. **k** ECC proteins reduced in the Jph2 IP from muscle of Speg-KO mice. **l** ECC proteins reduced in the Jph2 IP from the muscle of HA-Speg mice ($n = 6$–$7$). **m** Effect of Speg deficiency on the relative amounts of Esd, Cmya5 and Fsd2 in the Jph2 IP from Speg-KO mice. **n** Effect of Speg deficiency on the relative amounts of Esd, Cmya5 and Fsd2 in the Jph2 IP from HA-Speg mice. **o** Model summarizing the findings from the proteomic studies in skeletal muscle. Data are plotted as the mean ± SD. ****$P < 0.0001$, ***$P < 0.001$, **$P < 0.01$, and *$P < 0.05$.

dyadic complexes. A model of the dyadrsquo;s protein interactions is shown in Fig. 6l.

## Discussion

Excitation-contraction coupling is conducted by a highly complex assembly of conformationally coupled proteins localized to triads and dyads. The interactions of Speg and Jph2 and the consequences of Speg deficiency have been studied in cardiac[5,11,12], and skeletal muscle[10,14,15,39] but many questions especially about the targets of Speg both with respect to binding partners and phosphorylation targets in the dyads and triads remain to be fully defined. To elucidate the role of Speg in triad and dyad function, we created a mouse model of Speg deficiency (HA-Speg mice) that serves dual purposes of facilitating the identification of Speg-binding proteins and serving as a model of Speg deficiency with Spegβ and Spegα expression decreased by ~95% and 50%, respectively, in skeletal muscle of homozygous HA-Speg mice (Fig. 1a). In hearts of HA-Speg mice, Spegβ and Spegα levels were reduced by ~80% and 40%, respectively (Fig. 1c). The decrease in Spegβ protein levels in these mice likely reflects an effect of the inserted tag on both transcription (Spegβ mRNA decreased by ~50%) and increased turnover of the Spegβ protein. The 3XHA-Tag is known to alter protein stability[55]. Spegα is produced by transcription initiation at a second promoter site within the *Speg* gene and differs from Spegβ only in the N-terminus where it is missing 854 amino acids. Spegα is, therefore, not labeled by the HA-tag. This raises the question of why there is a decrease in Spegα in HA-Speg mice. We detect no differences in the mRNA for Spegα, suggesting that the decrease is more likely due to increased protein turnover. We find that Spegα forms hetero-oligomers with Spegβ, suggesting that complexed Spegα may be targeted for degradation. We propose that the HA-Speg mice are a mouse model of Spegβ deficiency while sparing Spegα to some degree, perhaps serving a mouse model of the relatively mild myopathy in humans associated with decreased Spegβ with sparing of Spegα[7]. In contrast, MCK-Cre⁺*Speg^{fl/fl}* mice[11] (which we designate as Speg-KO mice) display a >95% decrease in both Spegβ and Spegα in both skeletal muscle (Fig. 1b) and heart (Fig. 1d) and display a life-threatening myopathy where both skeletal and cardiac muscle are severely affected.

In comparing HA-Speg and Speg-KO mice, we found intriguing similarities and differences that shed light on the drivers of the more severe phenotype (weakness and atrophy in skeletal muscle and dilated cardiomyopathy) of Speg-KO mice. All muscles evaluated (EDL, soleus, and diaphragm in male and female mice) from Speg-KO mice displayed major reductions in ability to generate force and rightward shifts in the force-frequency curves. The most affected muscle in Speg-KO mice was the EDL. All three fiber types (IIa, IIx, and IIb) in the EDL show large decreases in CSA (Fig. 2h, l) with fiber-type distribution changes (Fig. 2l). In contrast, HA-Speg mice showed almost total sparing of the EDL muscle with respect to force generation, CSA, and changes in fiber-type distribution. The soleus and diaphragm muscles of HA-Speg mice have a decreased ability to generate force. In addition, the HA-Speg mice do not develop a dilated cardiomyopathy. These findings suggest that Spegα in HA-Speg mice can at least partially compensate for the loss of Spegβ in both skeletal and cardiac muscle and can completely rescue the EDL (the most affected muscle evaluated in the Speg-KO), suggesting that Spegα is critical in fast twitch muscles like the EDL.

In addition to reduced force generation, the skeletal muscle of Speg-KO mice displayed abnormal triads and altered Ca²⁺ handling and ECC[39]. Speg deficiency in cardiac tissue leads to heart failure (HF) with increased SR Ca²⁺ spark frequency and disruption of the t-tubules[12]. Because of the association of Speg deficiency with altered Ca²⁺ handling and t-tubule structure, we compared t-tubule structure and the frequency of spontaneous Ca²⁺ sparks in FDB fibers from control and Speg-KO mice and from control and HA-Speg mice (Fig. 3). In HA-Speg mice, we found small but significant increase in Ca²⁺ sparks and we found small areas of t-tubule disruption. Intriguingly, the Ca²⁺ sparks appeared to localize to these areas of t-tubule disruption (Fig. 3b). In contrast, the FDB fibers of Speg-KO mice displayed major areas of t-tubule disruption and a marked increase in spontaneous Ca²⁺ sparks. Again, the Ca²⁺ sparks in the fibers from Speg-KO mice primarily localize in areas of t-tubule disruption. The differences in the Ca²⁺ spark frequency and t-tubule disruption correlate with the relative severity of the muscle dysfunction in the HA-Speg versus Speg-KO mice. Our data suggest that both t-tubule disruption and Ca²⁺ leak are intricately linked phenomena associated with Speg deficiency and contribute to an amplifying feed-forward mechanism that leads to loss of muscle function. It is highly likely that both increased Ca²⁺ sparks and t-tubule disruption must occur to cause the severe cardiac and skeletal muscle functional losses in the Speg-KO mice. A contribution of decreased muscle satellite cells in the Speg-KO phenotype[40], especially the extensive EDL atrophy, is likely but requires further study.

Cardiac stress correlates with decreased Jph2[58–60]. In response to elevation in cytosolic Ca²⁺ levels, Jph2 undergoes calpain-mediated cleavage into fragments[49,50,61,62], and the N-terminal fragment moves to the nucleus to function as a stress-activated transcriptional repressor[50]. A C-terminal Jph2 fragment also translocates into the nucleus, where it is believed to affect hypertrophic signaling[48]. Calpain-mediated cleavage of Jph2 is a likely cause of t-tubule disruption[23,45,50,51,61,65]. Junctophilins are

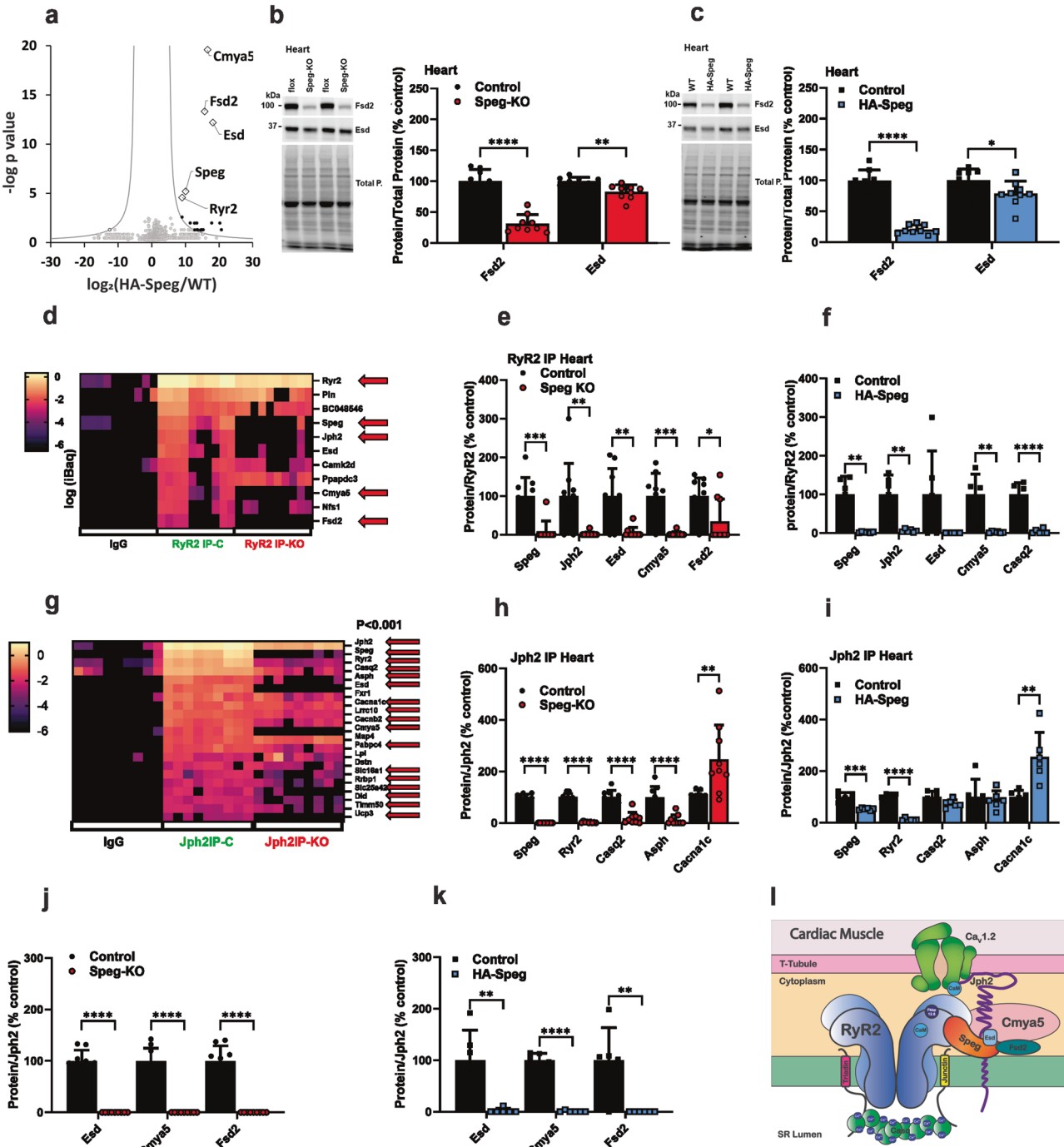

**Fig. 6 Speg-binding proteins and effects of Speg deficiency in the heart. a** Proteins immunoprecipitated from the hearts of HA-Speg mice using antibodies against the HA-tag and identified by mass spectrometry ($n = 6$ for each). **b** Effects of Speg deficiency on the levels of Fsd2 ($n = 8$ for each) and Esd ($n = 9$ for each) in homogenates from hearts of Speg-KO mice. A representative western blot is shown on the left. **c** Effects of Speg deficiency on the levels of Fsd2 and Esd in homogenates from hearts of HA-Speg mice ($n = 6$ for each). **d** Heatmap of specific RyR2 binding proteins from the hearts of control and Speg-KO mice ($n = 9$ for each). **e** Proteins that are significantly changed the cardiac RyR2 IP from controls compared Speg-KO mice. **f** Proteins that are significantly changed the cardiac RyR2 IP from controls compared HA-Speg mice ($n = 6$ for each). **g** Heatmap of specific Jph2-binding proteins from the hearts of control and Speg-KO mice ($n = 9$ for each). **h** Proteins that are significantly changed the cardiac Jph2 IPs from controls compared Speg-KO mice. **i** Proteins that are significantly changed the cardiac Jph2 IPs from controls compared HA-Speg mice ($n = 6$ for each). **j** Effect of Speg deficiency on the relative amounts of Esd, Cmya5 and Fsd2 in the cardiac Jph2 IP from Speg-KO mice. **k** Effect of Speg deficiency on the relative amounts of Esd, Cmya5, and Fsd2 in the cardiac Jph2 IP from HA-Speg mice. **l** Model summarizing the findings from the proteomic studies in cardiac muscle. Data are plotted as the mean ± SD. ****$P < 0.0001$, ***$P < 0.001$, **$P < 0.01$, and *$P < 0.05$.

required for maintenance of the structure of triads in skeletal muscle and dyads in cardiac muscle[63], but neither the N- nor C-terminal Jph2 fragments rescue ECC in Jph2-deficient cardiomyocytes[65], suggesting that the transmembrane domain at the C-terminus that anchors Jph2 to the SR membrane and the N-terminus that is involved with interactions with t-tubules are both required to prevent dyadic uncoupling and disruption of ECC. Both Speg-KO and HA-Speg mice displayed decreases in RyR1, $Ca_V1.1\alpha1$ (Cacna1s), and Jph2 in skeletal muscle, with the decreases being greater in the Speg-KO muscle. Neither RyR2 nor $Ca_V1.2\alpha1$ (Cacna1c) were decreased in the Speg-KO or HA-Speg hearts. Jph2, however, was decreased in both the HA-Speg and Seg-KO hearts with the extent of the decrease greater in Speg-KO mice. Hence, ECC proteins appear to be targeted for turnover in skeletal muscle but not in the heart by Speg deficiency. We examined the Jph2 fragmentation patterns in skeletal and cardiac muscle of controls, Speg-KO, and HA-Speg mice and found that sites at both the N-and the C-terminus of Jph2 were more accessible to cleavage in both heart and skeletal muscle of Speg-KO mice, but not HA-Speg mice (Fig. 4). These studies suggest that Spegα and Spegβ normally protects Jph2 from calpain cleavage, but, in HA-Speg mice, the presence of Spegα alone can protect Jph2 from this cleavage.

The primary sequence of Speg suggests that it interacts with other proteins and, also, as a dual kinase, phosphorylates multiple protein targets. Both RyRs[11] and Jph2[12] are phosphorylated by Speg. Whether the functional consequences of Speg deficiency in either skeletal muscle or heart arise from altered phosphorylation of Speg targets and/or from alterations in Speg interactions with other proteins is not yet known. The loss of phosphorylation of RyRs at the site of phosphorylation by Speg could alter open probability directly and/or decrease RyRs interaction with a modulatory protein that controls its opening. In addition, it is possible that RyR2 phosphorylation itself affects the binding of various interacting proteins[76]. Either mechanism could lead to SR $Ca^{2+}$ leak. Loss of Jph2 phosphorylation by Speg deficiency could alter its sensitivity to calpain cleavage and its ability to stabilize t-tubules. Alterations in Jph2 phosphorylation or its calpain cleavage could also alter its ability to interact with RyRs or voltage-dependent $Ca^{2+}$ channels. These possibilities argue for the need for more information about Speg interactions in both cardiac and skeletal muscle.

While HA-Spegβ expression levels are greatly decreased in the muscle of HA-Speg mice, HA-Spegβ is expressed at adequate levels to purify and identify Speg-binding proteins. To evaluate the effects of Speg deficiency on protein-protein interactions within triads and dyads, we used HA-Speg to immunoprecipitate Speg-binding proteins from skeletal muscle and heart and found that Esd, Cmya5, Fsd2, Jph1, and Jph2 were associated with Speg in skeletal muscle and Esd, Cmya5, Fsd2, Jph2, and RyR2 were associated with Speg in cardiac muscle. We confirmed these interactions using Fsd2 immunoprecipitation in skeletal muscle and showed that RyR1, Cmya5, Jph2, and Jph1 interactions with Fsd2 require Speg. We also showed that these proteins are localized to skeletal muscle triads using a BioID approach with Fkbp12-BirA. Finally, we immunoprecipitated RyR1 and Jph2 from homogenates of skeletal muscle and RyR2 and Jph2 from homogenates from the heart and showed that RyR1, RyR2 and Jph2 (both cardiac and skeletal muscle) interact with Cmya5, Fsd2 and Esd in a Speg-dependent manner. Determination of whether these proteins interact with low affinity with all triads/dyads or with high affinity to a subset of triads/dyads will require additional studies with different approaches, such as near-neighbor crosslinking and localization studies.

Another particularly important aspect of our studies is that Speg regulates the interactions of known components of triads and dyads. Speg deficiency decreases the interaction between Jph2 with RyR1 in skeletal muscle and between Jph2 and RyR2 in cardiac muscle, as shown in both RyR and Jph2 IPs. In skeletal muscle Jph2 IPs, Speg deficiency decreases the Jph2 interaction with $Ca_V1.1\alpha1s$ but in cardiac tissue Speg deficiency increases its interaction with $Ca_V1.2\alpha1c$. This difference may reflect differences in the rate at which the N-terminal Jph2 fragment dissociates from the voltage-dependent $Ca^{2+}$ channels to move it to the nucleus in the two tissues.

The presence of obscurin in the RyR1 and Jph2 IPs was unexpected, as was its loss in the IPs from Speg-KO mice. Obscurin deficiency impairs skeletal muscle $Ca^{2+}$ handling[77] and, hence, the loss of obscurin in the RyR and JPh2 IPs in Speg-KO mice could contribute to the phenotype. This finding raises the possibility that obscurin alters $Ca^{2+}$ handling in skeletal muscle via direct interactions within the triad. Unlike Cmya5, Esd Fsd2, Jph1, Jph2, and RyR1, obscurin is not found in the HA-Speg IPs from skeletal muscle, suggesting that its interaction is not directly with Speg but its presence in the triads may require an action (e.g., phosphorylation) or interaction of Speg.

Fsd2 (also called minispryn) and Cmya5 (also called myospryn) are paralogs that have previously been shown to interact with RyR2[75]. Immunoprecipitation of Fsd2 led to coimmuno-precipitation of ECC proteins, Cmya5 and Esd. Only the interaction of Esd with Fsd2 was not dependent on the presence of Speg, suggesting that Esd and Fsd2 form a complex. Cmya5 promotes RyR2 clustering in heterologous cells[75], plays a critical role in maintaining cardiac dyad architecture and positioning[43] and its ablation causes dyad disruption and SR $Ca^{2+}$ leak, leading to cardiac dysfunction[43]. Cmya5 is also considered to be a biomarker for several striated muscle diseases[78,79]. The third Speg-dependent component of the ECC complex is Esterase D, a widely expressed esterase with both carboxylesterase and thioesterase activity that plays critical roles in drug metabolism and suppression of cancer growth[80]. Our data strongly support a role for Speg bringing Esd, Fsd2, and Cmya5 to triads and dyads to stabilize these structures. One potential role for Speg, Esd, Fsd2, and Cmya5 in skeletal and cardiac muscle is to drive or maintain clustering of RyRs to stabilize triads and dyads. Such a role is consistent with the loss of interactions among ECC proteins in the absence of Speg.

A major question raised by our study is which of the differences between the HA-Speg and Speg-KO mice are the cause of the more severe phenotype of the Speg-KO mice. Cmya5, Fsd2 and Esd are lost from the ECC proteins of both triads and dyads in both mouse models and, hence, this loss alone cannot account for the Speg-KO phenotype. Since these same proteins are lost from triads and dyads in HA-Speg mice, their presence in triads and dyads must be dependent on Spegβ rather than Spegα (spared in HA-Speg mice). Speg-KO mice have increased $Ca^{2+}$ sparks, more t-tubule disruption, and display increased accumulation of calpain-mediated Jph2 fragments compared to HA-Speg mice, suggesting that Spegα may limit these events in the HA-Speg mice. In an inducible cardiac-specific Speg knockout mouse model, $Ca^{2+}$ sparks were not detected until after heart failure[12], suggesting that Speg deficiency does not necessarily directly lead to $Ca^{2+}$ sparks. Instead, Quick et al.[12] suggested that acute loss of Speg leads to decreased Jph2 phosphorylation and disruption of t-tubules.

Spegβ interacts with Esd, Fsd2, and Cmya5 and localizes these proteins to triads and dyads. The HA-Speg mice, while having a phenotype that is milder than the Speg-KO, display decreases in force generation in the soleus and diaphragm with almost total sparing of the EDL, suggesting that the phenotype may manifest primary in muscles that are constantly working. In this situation, the contributions of Cmya5, Esd, and Fsd2 to stabilization of the

triadic and dyadic ECC complex may play a more important role in muscle function. Our data suggest that, while t-tubule disruption may be a driver of the disease, the Speg-mediated mechanisms that drive t-tubule disruption are likely to be multifactorial and involve a series of closely coupled events including altered Jph2 phosphorylation[12], altered SERCA activity[13], increased $Ca^{2+}$ sparks, calpain-mediated cleavage of Jph2, and destabilization of interactions among the components of the triads and dyads due, at least partially, to the loss of Cmya5, Esd, and Fsd2.

## Methods

**Materials**. All antibodies used are listed in Supplementary Table 3.

**Mice**. The floxed Speg ($Speg^{fl/fl}$) mice were obtained ES cell clone EPD0180_2_A07 obtained from the KOMP repository (www.komp.com) and generated by the Wellcome Trust Sanger Institute. HA-Speg mice were created in the Genetically Engineered Rodent Model (GERM) Core at BCM. All mouse experiments complied with relevant ethical regulations for animal tested and were performed under IACUC protocol AN-2656.

**CRISPR-mediated knock-in allele design and reagent production for HA-Speg mice (Supplementary Fig. 1)**. To generate the *Speg-V5-3xHA* tag KI allele, a CRISPR-based targeting strategy was designed by the BCM Genetically Engineered Rodent Model Core (BCM GERM Core). The Core has experience in designing and producing mouse models[81–84]. Single-guide RNAs (gRNAs) were selected using the Wellcome Trust Sanger Institute Genome Editing website. This gRNA targets a double-strand break proximal to the initial methionine within the coding sequence of Speg (https://wge.stemcell.sanger.ac.uk/crispr/307086362) and was synthesized by Synthego (Redwood City, CA). The gRNA chosen had no predicted off-target sites with less than 3-base mismatches in exons or on the same chromosome. To introduce the V5-3xHA sequence into the N-terminus of *Speg*, a donor DNA template was synthesized by IDT (Coralville, IA) as a megamer. Single-guide RNA (20 ng/μL), Megamer (30 ng/μL), and Cas9 mRNA (100 ng/μl; Thermo Fisher Scientific, Waltham, MA) were mixed in a final volume of 50 μL 1x Modified TE (RNAse-free).

**Microinjection of CRISPR/Cas9 reagents**. C57BL/6J female mice, 24–32 days old, were injected with 5 IU/mouse of pregnant mare serum, followed 46.5 h later with 5 IU/mouse of human chorionic gonadotropin, and later mated to C57BL/6J males. Fertilized oocytes were collected at 0.5 dpc for microinjection. The BCM Genetically Engineered Mouse Core microinjected the sgRNA/donor/Cas9 mixture into the cytoplasm of at least 200 pronuclear stage zygotes. Injected zygotes were transferred into pseudo-pregnant ICR females on the afternoon of the injection, ~25–32 zygotes per recipient female.

**Genotyping**. Allele characterization was performed by standard PCR to identify founders with the inserted V5-3xHA tag, followed by Sanger sequencing. Three separate PCR reactions were performed, one reaction spanning the full insert (F 5'-CTCCACTGCTCATTCGCAAC, R 5'- CATCCCGGAACCAGCTGAG, F + R = 472 bp WT allele, 628 bp targeted allele) and two reactions to cover from outside each homology arm to the middle of the inserted sequence (5' arm F: 5'-CTCCACTGCTCATTCGCAAC-3', 5' arm R: 5-TAGCCCGCATAGTCAGGAAC-3', 314 bp targeted allele; 3' arm F: 5'-AGCCTATCCCTAACCCTCTCCT-3', 3' arm R: 5'-CTACTACCCGCAGCCTTACA-3', 356 bp). Mice with the appropriately sized band for the targeted allele and PCR

products for the two homology arm reactions were sequenced to verify the integrity of the targeted sequence. Targeted HA-Speg N0 founders were backcrossed with stock C57BL6/J animals to generate N1 pups. Heterozygous HA-Speg N1 mice had the same PCR assays performed, followed by sequence confirmation prior to establishing a colony from a limited number of N1 animals.

**Echocardiography**. Left ventricular function in mice was assessed by echocardiography using a VisualSonics Vevo F2 imaging system equipped with a frequency probe (40 MHz) in the BCM Mouse Metabolism & Phenotyping Core. After being anesthetized by 2% isoflurane mixed with 100% $O_2$, mice were placed on a heated pad where all four limbs were taped down onto surface electrodes to measure ECG. On the plate, anesthesia was maintained with 1.5% isoflurane. Body temperature was monitored using a rectal thermometer and kept between 36.5 °C and 37.5 °C. M-mode measurements were recorded at heart rates of $500 \pm 50$ bpm.

**Gene expression analysis**. Total RNA was extracted from mouse tissues using TRIzol reagent (Life Technologies) and purified using the RNeasy Fibrous Tissue Mini Kit (Qiagen). The purified RNA was reverse transcribed into cDNA using the SuperScript IV VILO Master Mix (Invitrogen). Gene expression assay was performed with the TaqMan Fast Advanced Master Mix (Applied Biosystems) using the following TaqMan probes: *Spegβ* (Mm00812485_m1), *Spegα and β* (Mm00812511_m1), *Apeg-1* and *Bpeg* (Mm00500180_m1). Using the qPCRBIO SyGreen Blue Mix Lo-ROX (PCR Biosystems), the following primers were used: *Spegα* F: 5'-TTCCTTTCCCTGCACATTCC-3' and *Spegα* R: 5'-TCTCCTGGGTGCTGTCTC-3'.

Samples were run in technical triplicates on the ViiA 7 Real-Time PCR System (Applied Biosystems). Mean Ct values for each gene were normalized against house-keeping genes *β-actin* (Mm02619580_g1) or *Gapdh* F: 5'-AGGTCGGTGTGAACGGATTTG-3' and R: 5'-TGTAGACCATGTAGTTGAGGTCA-3' to obtain ΔCt. ΔΔCt, which was calculated after normalizing to WT values, were then log2-transformed to obtain fold change values.

**Ex vivo force measurement**. As described previously[68,85], intact muscles were dissected and placed in a dish with fresh Krebs-Ringer solution (137 mM NaCl, 5 mM KCl, 1 mM $NaH_2PO_4$, 24 mM $NaHCO_3$, 5 mM glucose, 2 mM $CaCl_2$ and 1 mM $MgSO_4$) bubbled with a 5/95% mixture of $CO_2/O_2$. Muscles were tied and suspended between a force transducer and glass stationary anchor to equilibrate in a 34 °C bath. To get the optimal muscle length, a series of electrical stimulations for single twitches (train rate: 1 tps, train duration: 200 ms/train, stimulation duration: 0.2 ms/pulse, voltage: 20 V, resistance: 25 Ω) were applied to the muscles. Muscles were then adjusted to the length that allowed them to generate the strongest isometric tension. Force measurements were acquired by applying 250 ms trains using the frequencies 1 Hz, 15 Hz, 30 Hz, 40 Hz, 60 Hz, 80 Hz, 160 Hz, and 300 Hz, separated by a 1–2-min rest time. Muscle stimulation was applied using platinum electrodes attached to a Grass S48 stimulator, and muscle contracture was recorded by Chart5 (version 5.2).

**Sequential visualization of $Ca^{2+}$ sparks and t-tubules in living fibers**. As previously described[86], intact FDB muscles were dissected from mice and incubated in DMEM (Thermo Fisher Scientific) containing 0.1% penicillin-streptomycin (Thermo Fisher Scientific) and 0.4% collagenase A (Sigma) at 37 °C for 2.0 h. Following enzymatic digestion, isolated fibers were transferred to DMEM supplemented with 10% FBS and incubated in 5% $CO_2$ at 37 °C overnight. For measurement of spontaneous $Ca^{2+}$ sparks,

isolated FDB myofibers were loaded with the $Ca^{2+}$-sensitive dye Fluo-4AM (5 μM; Invitrogen) prepared in HEPES-Ringer solution (140 mM NaCl, 4.0 mM KCl, 1.0 mM $MgSO_4$, 5.0 mM $NaHCO_3$, 10.0 mM glucose, 10.0 mM HEPES, pH 7.3) for ~30 min at 37 °C and washed 3 times with dye-free HEPES-Ringer solution, followed by a rest period of 15 min to allow for dye de-esterification. T-tubules were visualized by incubating the fibers in the lipophilic styryl membrane marker FM*4-64 (6.5 mM, Thermo Fisher Scientific) for 10 min. High-resolution images were obtained on a Zeiss LSM 880 with a 40× NA1.2 water objective. The pixel dimensions were $0.06 \times 0.06$ μm in the x- and y axes. Fluo-4 was excited at 488 nm, and the emission intensity was captured between 491 and 556 nm in confocal mode. A sequence of 30 XY images was collected over time, and the frequency of sparks was determined from the Fluo-4 channel using custom routines written in IDL as previously described[87]. After Fluo-4 imaging, FM®4-64 was excited at 561 nm, and emission intensity was measured with an LP 650 nm in Super-resolution Airyscan mode in the same z-plane. The x and y locations of the spark centers were mapped onto the t-tubule image, and 100 μm² regions of interest centered at the center of the spark were extracted for processing of the t-tubule structure using AutoTT[88]. As control wild-type myofibers do not elicit $Ca^{2+}$ sparks, we used the spark centers obtained from Speg-deficient myofibers imaged the same day to select t-tubule regions of interest in a nonbiased fashion.

**Body composition**. Mice were anesthetized with isoflurane and assessed using the dual-energy x-ray absorptiometry (DEXA) system (Faxitron, Tucson, AZ, USA).

**Fiber typing and cross-sectional area**. Soleus and EDL muscles were dissected, embedded in OCT compound (Tissue-Tek), and frozen in 2-methylbutane. Frozen muscles were cut into 10 μm sections and stored at −20 °C. Frozen sections were prepared for fiber typing and experiments, as previously described[56].

**Immunoprecipitations**. Gastrocnemius muscle and heart from HA-Speg, Speg-deficient, and control mice were homogenized in six to eight volumes of IP buffer (w/v) (50 mM Tris-HCl, pH 7.4, 170 mM NaCl, 1 mM EDTA, and 1% NP-40) containing protease and phosphatase inhibitors using a bead homogenizer (Precellys® tissue homogenizer). After centrifugation, the cleared supernatant was incubated with primary antibodies and nonimmunized IgG for 1 h (for RyR) to overnight (for Jph2 and HA-tag). After incubation with primary Ab, 40 μl of magnetic beads (Dynabeads® Protein G, Invitrogen) were added and incubated for 1 h at 4 °C. Magnetic beads were washed three times with IP buffer and twice with sterile PBS using a magnetic stand. Washed beads were trypsinized for mass spectrometry.

**SDS-PAGE and western blot**. Skeletal muscle and heart were snap-frozen in liquid nitrogen and stored at −80 °C until use. Skeletal muscle and heart were thawed and homogenized in RIPA buffer (150 mM NaCl, 1% NP-40, 0.5% sodium deoxycholate, 0.1% SDS, and 50 mM Tris-HCl, pH 8.0) supplemented with a protease and phosphatase inhibitor cocktail by using a bead homogenizer (Precellys 24 homogenizer, Bertin Instruments, Paris, France). Protein concentrations were determined using a BCA assay. Homogenates (60 μg) were separated by SDS-PAGE using the Bio-Rad Stain Free gel system to enable total protein visualization and quantitation. Proteins on PAGE gels were transferred to PVDF membranes (Immobilon-FL, EMD Millipore), blocked in EveryBlot blocking buffer (Bio-Rad), and incubated with primary antibodies overnight at 4 °C.

After washing in PBS containing 0.2% Tween-20 (PBST), membranes were incubated with fluorophore-tagged Starbright secondary antibodies (Bio-Rad) for 1 h at room temperature. After washing in PBST, blots were imaged on a Chemidoc MP system (Bio-Rad). Total protein using stain-free gels (Bio-Rad), after transfer of the proteins to the PVDF membrane (Immobilon®-FL, Millipore), the PVDF membrane was imaged by using a ChemiDoc™ MP imaging system (Bio-Rad). All antibodies used are provided in Supplementary Table 3.

**BioID constructs**. All BioID constructs used here were based on pENTRY-EF1a (Invitrogen). The coding sequences of FKBP12 or GFP were fused with that of BirA*, followed by P2A- acGFP in pENTRY-EF1a. P2A-AcGFP was added to mark transfected cells (Supplementary Fig. 6A, B).

**AAV production**. AAV transgene plasmids (Supplementary Fig. 5) containing AAV2 inverted terminal repeats (ITRs) were constructed using gene synthesis and standard molecular biology techniques. Plasmid "1533_pAAV-CB-AcGFP-BirA-HA" expresses an *Aequorea coerulescens* GFP (AcGFP) with a Gly-Gly-Ser-Gly flexible linker fused to a C-terminal BirA* biotin ligase followed by an Ala and a Hemagglutinin (HA) epitope tag. The transgene cassette is driven by a chicken beta-actin promoter with a CMV enhancer element and includes a 3' woodchuck hepatitis virus posttranscriptional regulatory element (WPRE) and bovine growth hormone polyadenylation signal downstream. A plasmid in the identical configuration named "1529_pAAV-CB-FKBP12-BirA-HA" was generated to express the murine FKBP12-BirA-HA fusion protein. AAV vectors based on serotype 9 were generated by the triple transfection method of Xiao et al.[89] in HEK293T cells with minor modifications[90]. AAV9 was purified by CsCl density gradient ultracentrifugation, dialyzed, concentrated in phosphate-buffered saline, aliquoted, and stored at −80 °C until use. Viral titers were determined by qPCR with SYBR green compared to a standard curve from tenfold serial dilutions of the respective transgene plasmid. Primers specific to the WPRE were used: AH_0104—5'-CATTGCCACCACCTGTCAGC-3'; AH_0105—5'GACGTAGCAGAAGGACGTCC-3'.

**Mass spectrometry**. Immunoaffinity purification followed by nanoHPLC-MS/MS analysis was used for interactome as previously reported with minor modifications[91]. Briefly, the affinity-purified protein and interacting proteins are digested on beads using Trypsin/Lys-C enzyme, then the digested peptide is enriched and desalted by in-housed STAGE tip[92] column with 2 mg of C18 beads (3 μm, Dr. Maisch GmbH, Germany) and vacuum dried. Resuspended peptides were analyzed on an nLC 1000 coupled with an Orbitrap Fusion mass spectrometer (Thermo Scientific) with an ESI source. Data acquisition was performed in data-dependent analysis mode (DDA) for unbiased peptide detection. The obtained spectra were processed by the Proteome Discoverer 2.1 interface (PD 2.5, Thermo Fisher) with the Mascot algorithm (Mascot 2.1, Matrix Science). Assigned peptides were filtered with a 1% false discovery rate (FDR) using percolator validation based on the q-value. The calculated area under the curve of peptides was used to calculate the iBAQ for protein abundance using in-house software[93].

**Statistics and reproducibility**. All samples analyzed were distinct samples. For $Ca^{2+}$ imaging, multiple FDB fibers from each mouse were analyzed but we used nested analyses of fibers from at least three mice of each genotype. Differences between means of two groups were analyzed for significance using Student's *t* tests and normality tests. If the data failed the normality tests,

we performed a Mann–Whitney rank-sum test. Differences between means of multiple groups were analyzed by one- or two-way ANOVA, and those with multiple individual myofibers per subject were analyzed by nested (hierarchical) ANOVA with Tukey post hoc tests. Mean differences between groups for longitudinal measurements as a function of age are analyzed by repeated measure ANOVA with Tukey post hoc tests. Values of $P < 0.05$ (95% confidence) were considered statistically significant (unless otherwise indicated). Immunoprecipitations were first analyzed for specificity by comparing proteins in IgG controls to those in RyR1 or Jph2 IPs. We used a criterion of >20× purification and $P < 0.01$. These values were used in the heat maps. However, for discovery proteomic data were analyzed with Holm–Šídáks multiple comparisons tests and/or calculation of the false discovery rate (FDR)[94]. We also used a targeted approach for comparison of the effects of Speg deficiency on known ECC proteins. For comparison of the effects of Speg deficiency the proteins were normalized to the amount of either RyR1 or Jph2 in the IPs and then batch-corrected to the average of the controls samples in each separate experiment. These values are plotted and compared as % control and analyzed by Student's $t$ tests and normality tests.

**Reporting summary**. Further information on research design is available in the Nature Portfolio Reporting Summary linked to this article.

## Data availability

Complete datasets from all proteomic studies (Supplementary Data 1) and for Figs. 1–6 (Supplementary Data 2) are provided as supplementary data. All data were generated in the laboratories of Drs. Hamilton, Sung and Rodney. All data in the Supplementary Data Tables and Figs. are in Supplementary Data 3, and all uncropped gels are provided in Supplementary Data 4. The MS data (Supplementary Data 1) have been deposited to the ProteomeXchange Consortium (https://proteomecentral.proteomexchange.org/) via the MASSIVE repository (MSV000092796) with the dataset identifier PXD044979. The plasmids for AAV-FKBP12-BirA and AAV-GFP-BirA (complete vector maps in Supplementary Fig. 6) are publicly available through Addgene (ID# 208200 and 208201).

## Code availability

Custom routines written in IDL will be made available upon request to George Rodney (Rodney@bcm.edu).

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

## Acknowledgements
We thank the directors and staff at the Genetically Engineered Rat and Mouse Core, the Mouse Metabolism and Phenotyping Core, the Optical Imaging and Vital Microscopy Cores at BCM and the Genetically Engineered Rodent Model (GERM) Core at BCM. We would also like to thank Dr. Robert Dirksen for sharing the design of the vector for the V5/HA-tag, and Drs. Ang Guo and Long-Sheng Song (University of Iowa Carver College) for supplying the AutoTT program. High-resolution X-ray microtomography was conducted in the Optical Imaging and Vital Microscopy Core at BCM, and echocardiography was conducted at the Mouse Metabolism and Phenotyping Core at Baylor College of Medicine. The Mouse Metabolism and Phenotyping Core is supported by RO1DK114356, UM1HG006348, and S10OD032380. The GERM Core is funded in part by the National Institutes of Health Cancer Center Grant (P30 CA125123). Research reported in this publication was supported by R01AR072475 to S.L.H., R01 HL132840 and R01 DK124477 to W.R.L., R01-HL089598, R01-147108, R01-HL153350 to X.H.T.W., and RO1AR061370 NIH/NIAMS to G.G.R.

## Author contributions
All authors helped with the writing of the manuscript, read, critiqued, and corrected multiple versions of the manuscript. and approved the last version. C.S.L. performed and analyzed data from western blotting, Jph2 fragmentation, and immunoprecipitations, helped supervise students, and performed the multiple aspects of the study. S.Y.J. performed and analyzed proteomic data, R.S.Z.Y. performed and analyzed in vivo analyses of growth and body composition and analyzed mRNA levels for Speg isoforms, N.H.A. collected and analyzed t-tubule structure in FDB fibers and handled the mouse matings, J.H. performed and analyzed the BioID experiments, T.C. performed and analyzed cardiac echoes, L.B. performed and analyzed force-frequency studies and CSA/fiber-type distribution analyses, J.D.F. performed and analyzed Jph2 fragmentation studies, B.C. performed and analyzed force-frequency data, A.D.H. performed and analyzed force-frequency and echo data, C.S.W. supervised all in vivo mouse studies and optimized the echoes, D.L. created HA-Speg mice, A.E.H. prepared the AAV for the BioID experiments, P.Z. created the BioID targeting vectors, X.H.T.W. suggested experiments for the revision, critiqued the ms, and helped write the revision, WRL designed, created, and supervised the AAV vector studies, G.G.R. helped plan the study, supervised the force experiments and performed and analyzed the Ca$^{2+}$ sparks and t-tubule studies, S.L.H. supervised the entire study, planned the experiments, analyzed data, created the figures, and wrote the first and last versions of the manuscript.

## Competing interests
The authors declare the following competing interests: X.H.T.W. is a founding partner and board member of Elex Biotech, a start-up company that developed drug molecules to target ryanodine receptors to treat cardiac arrhythmias. The remaining authors declare no competing interests.
