## [Peer Review File · Communications Biology]

Reviewers' comments:

Reviewer #1 (Remarks to the Author):

The paper describes the relationship of SPEG with other triad/dyad proteins, its role in phosphorylating RYR1 and relative roles of SPEGa vs SPEGb. The paper is an example of incremental but important advancement in understanding the function of SPEG protein. I have no additional comments, it's a very nice paper well written.

Reviewer #2 (Remarks to the Author):

The authors address the role of Speg as a key regulator of the structural and functional interactions that govern excitation-contraction coupling and t-tubule structure, both in skeletal and cardiac muscle. They present solid evidence showing that the absence of Speg disrupts the protein complex engaged in excitation-contraction coupling (ECC) in both muscle types. In addition, they identify novel Speg binding proteins. These findings, which significantly advance our current knowledge on the proteins that are important for the ECC process, are of clear interest to the researchers who study skeletal and cardiac muscle function.

In addition, the manuscript presents a detailed description of the experimental procedures used by the authors, and the statistical analysis of results is adequate.

Minor points

A scheme showing all the new Speg binding proteins identified in this work in skeletal and cardiac muscle would be helpful.

Please define all abbreviations used the first time they are mentioned in the text. As an example, in the Introduction section MTM1, DNM2, BIN1, CACNA1S and TTN should be defined.

In the legend to figure 3H, please define the labels of the y-axis.

Some minor language corrections are required. For example, jJph2 should read Jph2 in the sentence "Another likely target of Speg phosphorylation is jJph2". Likewise, the sentence "Huntoon et al39 created striated muscle specific Speg-KO mice, developed a dilated cardiomyopathy.." would read better as "Huntoon et al39 created striated muscle specific Speg-KO mice, which developed a dilated cardiomyopathy.."

Reviewer #3 (Remarks to the Author):

The manuscript aims to elucidate the roles of SPEG in skeletal and cardiac muscle excitation-contraction coupling. The authors use two mouse models and muscle physiological measurements combined with proteomics analyses for their investigations. Despite the considerable amount of work presented in the manuscript, I have concerns about methodology and experimental design that complicates the interpretation of the data as outlined below.

Major points:

1. The authors employ CRISPR/Cas9 technology to add a V5/HA tag to the N-terminus of endogenous

SPEG in mice. Unexpectedly, this causes a marked decrease of SPEG- β (~95% and 82%, respectively) and a substantial reduction of SPEG α (~51% and 38%, respectively) in skeletal muscle and heart. Based on these data, the authors claim the HA-SPEG mouse is a model of SPEG β deficiency. However, this simplified interpretation neglects possible gene dose effects. To make a strong point of SPEG β isoform specific function, the authors should generate a SPEG β isoform specific knockout mouse model. The SPEG α and SPEG β differ at their N-terminus due to alternative splicing. It is not clear how the knockin HA affects the transcription of these two isoforms and whether the HA tagging might also affect their protein stability. Moreover, *Speg* locus also produces two other transcripts, *Apeg* and *Bpeg*. It is also worthy to find out whether and how the HA tagging affect the expression of *Apeg* and *Bpeg*.

2. In the HA-SPEG β proteomics study, the authors do not find some known SPEG-interactors such as SERCA, myotubularin and desmin, which raise a question whether the HA-SPEG β behaves in a manner similar to the endogenous SPEG. Is it possible that the HA-SPEG β in certain subcellular compartments is more prone for degradation while it is more stable in other subcellular compartments? The authors claim that they have identified three new SPEG interacting proteins, *Esd*, *Fsd2* and *Cmya5* (Page 6 line 2-3). They need to verify these interactions by co-immunoprecipitation of co-expressed recombinant *Speg* with these proteins in cells, and also perform co-immunoprecipitation with endogenous proteins.

3. The authors observe an increase of Ca²⁺ sparks in HA-SPEG mouse muscle fibers, which they ascribe to the hypophosphorylation of Ryr1 on Ser2902. This is somehow difficult to understand. Ryr1 protein is markedly decreased in HA-SPEG or SPEG-KO mouse muscle, which might lead to a decrease of non-phosphorylated Ryr1. Assumably, the non-phosphorylated Ryr1 underlies the Ca²⁺ leakage that results in the increased Ca²⁺ sparks. Therefore, the authors need to provide such evidence to show that non-phosphorylated Ryr1 is increased in HA-SPEG or SPEG-KO mouse muscle.

4. The Ser2902 phosphorylation of Ryr1 is purely based on the quantification of mass-spec data. There is not enough information about the methodology how the quantification is performed and what is the STD peptide. The authors need to verify this result using an independent approach, ideally with a site-specific phospho-antibody.

5. The claim that Ryr1 is phosphorylated by SPEG on its Ser2902 is rather weak. At least two experiments need to be done in order to draw such a conclusion. One is to co-express Ryr1 with SPEG in cells and then to detect Ryr1-S2902 phosphorylation on immunoprecipitated Ryr1. The other is to perform an in vitro phosphorylation of Ryr1 using recombinant SPEG.

6. It has been shown that SPEG plays a critical role in muscle development. In both mouse models used in the study, *Speg* expression would be affected in satellite cells, which might impact on muscle development. The increased Ca²⁺ sparks might be secondary to the impaired T-tubule integrity. In line with such a possibility, Ca²⁺ sparks remain normal in an inducible cardiac-specific *Speg* knockout mouse model when heart failure does not occur (PMID: 30566039, 27729468), but is increased after heart failure (PMID: 27729468). A better model to investigate this is to utilize an inducible muscle knockout model.

Minor points:

1. Gender difference is frequently observed in congenital myopathies. Therefore, it is important to specify the gender of mice used in the study. And also it is important not to use mixed gender in the experiments.

We thank the three reviewers for their insightful comments. We have made many changes in the ms to improve readability and impact.

To summarize our changes:

1. The manuscript has been extensively rewritten and edited to improve readability.
2. We have removed the phosphorylation studies to allow us to add additional proteomic data and to provide time to perform the additional phosphorylation studies suggested by reviewer 3. Since the phosphorylation data was not a major part of the previous version, we do not feel its removal alters the impact of the paper.
3. We added a 3 new proteomic datasets including RyR IPs from control and HA-Speg skeletal muscle and heart and Fsd2 IPs. We have also added heat maps of specific binding proteins for RyR1, RyR2, Fsd2, and Jph2 from skeletal muscle and heart.
4. We reorganized Fig. 4, 5, and 6 to make them easier to follow.
5. We have added new data in the functional figure (now Figure 2) not only to emphasize differences between HA-Speg and Speg KO mice, but also to present data on the Speg KO mice, that to our knowledge, have not previously been shown.
6. For the Jph2 fragmentation (now Fig. 4), we have added colored panels to indicate the use of two different antibodies and we added a fragmentation map to show Speg protected cleavage sites and location of antibody binding in Figure 4.
7. Immunoprecipitations from skeletal muscle and cardiac muscle are now separated into 2 figures (Fig. 5 and 6) to emphasize both the similarities and the differences. We have also added models to these figures.
8. New data added to this version: **1)** RyR IPs from skeletal muscle and heart of the HA-Speg mice; **2)** mRNA analyses for Speg β , Speg α and Apeg+Bpeg in the muscle of HA-Speg mice, **3)** Demonstration that HA-Speg IPs contain a significant amount of Speg α , suggesting the Speg β and Speg α are able to form heterooligomers, **4)** Data on the effects of Speg deficiencies in both models on both CSA and fiber type distribution in different muscles; **5)** demonstration that immunoprecipitation of Esd and Fsd2 coimmunoprecipitates Speg, supporting our hypothesis that these are Speg binding proteins. We were unable to find a good antibody for Cmya5 but studies in other laboratories have shown the interaction of Cmya5 in cardiac dyads (PMID 35449169, PMID: 28740084) **6)** Exciting new mass spec data from the immunoprecipitation of Fsd2 to show the coimmunoprecipitation of Esd, Cmya5, RyR1, Jph1 and Jph2. **7)** We have also added data to show the effects of Speg deficiencies in the two mouse models on the protein levels of Esd and Fsd2 in both skeletal muscle and heart.
9. Reviewer 3 critiqued our study and referred frequently to work from Xander Wehrens' lab. We, therefore, asked Dr. Wehrens for his help addressing some of the issues raised by reviewer 3. Dr Wehrens, a world renown expert in the role of Speg in heart, is a member of our department. Dr. Wehrens critically evaluated our data, made valuable suggestions for additional experiments, and helped to write the revision. We have, therefore, added him as a coauthor.
10. We have removed Dr. Filip Van Petegem from the list of authors (with his complete agreement) because his contributions related entirely to modeling the effects of phosphorylation of RyR1 at S2902 on RyR1 structure. He will be an author on a follow up paper that we are planning on phosphorylation of RyR1 and Jph2.

Reviewers' comments:

Reviewer #1 (Remarks to the Author):

The paper describes the relationship of SPEG with other triad/dyad proteins, its role in phosphorylating RYR1 and relative roles of SPEGa vs SPEGb. The paper is an example of incremental but important advancement in understanding the function of SPEG protein. I have no additional comments, it's a very nice paper well written.

Response. We thank the reviewer for the support of the study. After re-reading the previous version of this manuscript, we realized that we did not emphasize the advances. We have rewritten sections of the paper to emphasize the new findings of this study that we feel advance and expand our knowledge of the role of Speg. We would like to emphasize that all of the molecular changes that occur in the Speg KO, also occur in the HA-Speg (just different levels of changes), establishing HA-Speg mice as a new model of Speg deficiency with a less severe phenotype. This will eventually allow us to conduct aging studies on the role of Speg since the HA-Speg mice do not die prematurely.

Reviewer #2 (Remarks to the Author):

The authors address the role of Speg as a key regulator of the structural and functional interactions that govern excitation-contraction coupling and t-tubule structure, both in skeletal and cardiac muscle. They present solid evidence showing that the absence of Speg disrupts the protein complex engaged in excitation-contraction coupling (ECC) in both muscle types. In addition, they identify novel Speg binding proteins. These findings, which significantly advance our current knowledge on the proteins that are important for the ECC process, are of clear interest to the researchers who study skeletal and cardiac muscle function. In addition, the manuscript presents a detailed description of the experimental procedures used by the authors, and the statistical analysis of results is adequate.

Response. We thank reviewer 2 for his/her support of the manuscript and our findings.

Minor points

A scheme showing all the new Speg binding proteins identified in this work in skeletal and cardiac muscle would be helpful.

Response. We added models to both proteomic figures and we thank the reviewer for this excellent suggestion.

Please define all abbreviations used the first time they are mentioned in the text. As an example, in the Introduction section MTM1, DNM2, BIN1, CACNA1S and TTN should be defined. In the legend to figure 3H, please define the labels of the y-axis.

Response. We defined the abbreviations and defined the labels for Figure 3H in the figure legend and apologize for the oversight.

Some minor language corrections are required. For example, jJph2 should read Jph2 in the sentence “Another likely target of Speg phosphorylation is jJph2”. Likewise, the sentence “Huntoon et al39 created striated muscle specific Speg-KO mice, developed a dilated cardiomyopathy.” would read better as “Huntoon et al39 created striated muscle specific Speg-KO mice, which developed a dilated cardiomyopathy.”

Response: These corrections have been made.

Reviewer #3 (Remarks to the Author):

The manuscript aims to elucidate the roles of SPEG in skeletal and cardiac muscle excitation-contraction coupling. The authors use two mouse models and muscle physiological measurements combined with proteomics analyses for their investigations. Despite the considerable amount of work presented in the manuscript, I have concerns about methodology and experimental design that complicates the interpretation of the data as outlined below.

Major points:

1a. The authors employ CRISPR/Cas9 technology to add a V5/HA tag to the N-terminus of endogenous SPEG in mice. Unexpectedly, this causes a marked decrease of SPEG- β (~95% and 82%, respectively) and a substantial reduction of SPEG α (~51% and 38%, respectively) in skeletal muscle and heart. Based on these data, the authors claim the HA-SPEG mouse is a model of SPEG deficiency. However, this simplified interpretation neglects possible gene dose effects. To make a strong point of SPEG β isoform specific function, the authors should generate a SPEG β isoform specific knockout mouse model.

Response. We are confused about what exactly the reviewer means by gene dosage effects. We are working with mice homozygous for a targeted insertion of a sequence encoding the HA-tag into the Speg gene. Speg α and Speg β differ only in the N-terminus and we are inserting the sequence encoding the tag into exon 1 of the *Speg* gene, thereby adding a tag to only Speg β . The entire sequence for Speg α is contained within Speg β protein. These mice have been backcrossed multiple times to eliminate off target insertion of the HA-Tag. While we agree that a mouse model of Speg β only knockout would be of interest, based on our findings with the HA-Speg, these mice would likely have a very mild phenotype and are unlikely to tell us a lot more than the HA-Speg mice. It would also be difficult and time consuming to create this model (especially without altering Speg α expression). Based on our analyses that show that the HA-Speg mice undergo all of the same changes (smaller in magnitude) as the Speg-KO, we think the HA-Speg is a good model of a less severe Speg deficiency.

1b. “how the knockin HA affects the transcription of these two isoforms and whether the HA tagging might also affect their protein stability. Moreover, Speg locus also produces two other transcripts, Apeg and Bpeg. It is also worthy to find out whether and how the HA tagging affect the expression of Apeg and Bpeg.”

Response. The 3XHA-tag has previously been shown to decrease protein stability and expression of a variety of proteins and, hence, it is not surprising that it alters HA-Speg β protein levels (Vecchio et al, Mol Cell Neurosci, 2014, Saiz-Baggetto et al, Plos One, August 2017)).

These workers have suggested that the 3 X HA-tag destabilizes the tagged proteins and targets them for degradation. If HA-Speg β and Speg α form hetero-oligomers, the decrease in Speg α might be explained by degradation of these hetero-oligomers. In response to the reviewers comments we have performed rt-PCR for Speg β , Speg α , Apeg+Bpeg and show only a decrease in the message for Speg β in the HA-Speg homozygous mice. We have also added a discussion of the possible reasons for Speg β and Speg α decreases in the HA-Speg mice.

2. In the HA-SPEG β proteomics study, the authors do not find some known SPEG-interactors such as SERCA, myotubularin and desmin, which raise a question whether the HA-SPEG β behaves in a manner similar to the endogenous SPEG. Is it possible that the HA-SPEG β in certain subcellular compartments is more prone for degradation while it is more stable in other subcellular compartments?

Response. The reviewer raises an interesting question. We have added a discussion of four possible reasons that we do not detect the previously identified proteins, but the most obvious is that these are Speg α binding proteins. Since our goal was to identify Speg binding proteins in dyads and triads, degradation in other compartments with sparing in the dyads and triads would certainly be serendipitously beneficial to our study.

3. The authors claim that they have identified three new SPEG interacting proteins, Esd, Fsd2 and Cmya5 (Page 6 line 2-3). They need to verify these interactions by co-immunoprecipitation of co-expressed recombinant Speg with these proteins in cells, and also perform co-immunoprecipitation with endogenous proteins.

Response. We thank the reviewer for the suggestion. We have added Esd and Fsd2 IPs with western blots that demonstrate Speg, Esd, and Fsd2 are in both types of IP. More importantly, we have added mass spec data from an Fsd2 IP that shows that RyR1, Jph1, Jph2, Cmya3 and Esd co-immunoprecipitate with Fsd2.

3. The authors observe an increase in Ca²⁺ sparks in HA-Speg mouse muscle fibers, which they ascribe to the hypo-phosphorylation of Ryr1 on Ser2902. This is somehow difficult to understand. Ryr1 protein is markedly decreased in HA-SPEG or SPEG-KO mouse muscle, which might lead to a decrease of non-phosphorylated Ryr1. Assumably, the non-phosphorylated Ryr1 underlies the Ca²⁺ leakage that results in the increased Ca²⁺ sparks. Therefore, the authors need to provide such evidence to show that non-phosphorylated Ryr1 is increased in HA-SPEG or SPEG-KO mouse muscle.

Response. If phosphorylation stabilized a closed state of the channel, a decrease in this phosphorylation event could drive RyR1 Ca²⁺ leak. We agree with the reviewer and we do not yet have the data to fully support this. We have removed the discussion of RyR1 phosphorylation at S2902D from the paper and will pursue it further as suggested by the reviewer. We do not know the primary cause of SR Ca²⁺ leak (t-tubule disruption versus changes in RyR1 phosphorylation and/or RyR1 interactions that stabilize the closed state of the channel). These possibilities are raised in the discussion without a discussion of a specific site for phosphorylation.

4. The Ser2902 phosphorylation of Ryr1 is purely based on the quantification of mass-spec data. There is not enough information about the methodology how the quantification is performed and

what is the STD peptide. The authors need to verify this result using an independent approach, i site-specific phospho-antibody ideally with a site-specific phospho-antibody.

5. The claim that Ryr1 is phosphorylated by SPEG on its Ser2902 is rather weak. At least two experiments need to be done in order to draw such a conclusion. One is to co-express Ryr1 with SPEG in cells and then to detect Ryr1-S2902 phosphorylation on immunoprecipitated Ryr1. The other is to perform an in vitro phosphorylation of Ryr1 using recombinant SPEG.

Response to 4 and 5. We removed the RyR1 phosphorylation data. We agree with the reviewer about the need for additional studies on the phosphorylation and we thank him/her for the excellent suggestions. However, it should be noted that we provide a comprehensive analysis of ECC protein changes and changes in Speg, RyR and Jph2 binding proteins in the Speg deficient mice. These data are also needed to interpret phosphorylation data. Loss of Speg binding or phosphorylation of a site on RyR1 could also cause loss of an RyR1 modulator that holds the channel closed. We feel that it is important to determine if the functional outcome is due to the altered phosphorylation, the loss of RyR modulators or both. Our study is the first comprehensive and quantitative proteomic analyses of the effects of Speg deficiency in both skeletal and cardiac muscles and identifies a number of proteins that no longer appear to associate with triadic and dyadic ECC complexes.

6. It has been shown that Speg plays a critical role in muscle development. In both mouse models used in the study, Speg expression would be affected in satellite cells, which might impact on muscle development. The increased Ca^{2+} sparks might be secondary to the impaired T-tubule integrity. In line with such a possibility, Ca^{2+} sparks remain normal in an inducible cardiac-specific Speg knockout mouse model when heart failure does not occur (PMID: 30566039, 27729468), but is increased after heart failure (PMID: 27729468). A better model to investigate this is to utilize an inducible muscle knockout model.

Response. We thank the reviewer for raising this issue that, although previously mentioned, we did not adequately address. It is certainly possible that defective or decreased satellite cells contribute to the phenotype of the Speg-KO mice. We now have expanded our discussion to allow for this possibility. In the previous version and the current one, we discussed the potential roles of t-tubule disruption and Ca^{2+} sparks. We demonstrate that Ca^{2+} sparks occur mainly in regions of t-tubule disruption. Both Ca^{2+} sparks and t-tubule disruption are dramatically reduced in HA-Speg compared to Speg KO mice, which is consistent with both Ca^{2+} sparks and t-tubule disruption playing critical roles in the phenotype. The Cre recombinase under the MCK promoter turns on at birth and hence, is not likely to be playing a role in embryonic heart development. HA-Speg mice, despite showing many of the same molecular changes as Speg KO mice, do not display a cardiac phenotype and, hence, our studies basically agree with the above cited paper. HA-Speg (which should be turned on at the time the Speg gene is normally turned on) do not display a severe muscle force loss, or changes in CSA or fiber type distribution. We cannot say which comes first Ca^{2+} leak or t-tubule disruption but it is likely to be a cyclic process where Ca^{2+} leak causes t-tubule disruption and t-tubule disruption causes SR Ca^{2+} leak.

Minor points:

1. Gender difference is frequently observed in congenital myopathies. Therefore, it is important

to specify the gender of mice used in the study. And also it is important not to use mixed gender in the experiments.

Response. We **never** mix data from mice of different sexes **in any experiment**. However, we realize that our presentation of this might have been confusing. The sex of the mice is now indicated in all figure legends. We have performed all functional studies with both sexes separately and find that both sexes are affected by Speg deficiency. However, given the expense of the experiments, the proteomic data were generated primarily with male mice.

In summary, this paper includes a comprehensive comparison of both cardiac and skeletal muscle function in two models of Speg deficiency and evaluates the effects of Speg deficiencies on Ca^{2+} sparks, t-tubule structure, expression of ECC proteins, and Jph2 fragmentation. In addition, we provide a quantitative analysis of HA-Speg binding proteins, demonstrate that Speg is a near neighbor of Fkbp12 in the triad and identify new potential Speg, RyR and Jph2 binding proteins in heart and skeletal muscle and show how these interactions are altered by Speg deficiency. In summary, our findings advance our knowledge of the role of Speg, clarify the steps that drive disease severity and identify its novel binding partners in cardiac and skeletal muscle.

REVIEWERS' COMMENTS:

Reviewer #2 (Remarks to the Author):

The authors have answered satisfactorily my previous comments.

Reviewer #3 (Remarks to the Author):

My questions have been adequately addressed in the revised MS. I have no further question.